# Occupational Stress and Turnover Intentions in Employees of the Portuguese Tax and Customs Authority: Mediating Effect of Burnout and Moderating Effect of Motivation

Mariana Freitas [1], Ana Moreira [1,2,3,*] and Fernando Ramos [4]

1 School of Psychology, ISPA—Instituto Universitário, Rua do Jardim do Tabaco 34, 1149-041 Lisboa, Portugal; 27227@alunos.ispa.pt

2 APPsyCI—Applied Psychology Research Center Capabilities & Inclusion, ISPA—Instituto Universitário, R. do Jardim do Tabaco 34, 1149-041 Lisbon, Portugal

3 Faculdade de Ciências e Tecnologia, Universidade Europeia, Quinta do Bom Nome, Estr. da Correia 53, 1500-210 Lisboa, Portugal

4 APIT—Associação Sindical dos Profissionais da Inspeção Tributária e Aduaneira, Rua Duque da Terceira, 403, 1.º Esq, 4000-537 Porto, Portugal; frms62@gmail.com

* Correspondence: amoreira@ispa.pt

**Abstract:** The main objective of this study was to study the effect of occupational stress on turnover intentions, as well as determine whether this relationship is mediated by burnout levels and moderated by motivation. The sample size of this study consisted of 603 participants, all employees of the Portuguese Tax and Customs Authority, which is part of the Ministry of Finance and belongs to the Direct State Administration. The results indicate that occupational stress (with managers, career and pay, and overwork) has a positive and significant effect on turnover intentions and that this relationship is mediated by burnout. Motivation (intrinsic and identified) has a negative and significant effect on turnover intentions. Intrinsic motivation moderates the relationship between occupational stress (with managers, colleagues, career and pay, and family problems) and turnover intentions. These results indicate that among the dimensions of occupational stress, the most critical are the stress caused by managers, work overload, and career and pay. These are factors that the Direct State Administration should be concerned with to reduce employees' stress levels, as well as their turnover intentions.

**Keywords:** occupational stress; burnout; turnover intentions; motivation; quantitative study

## 1. Introduction

In recent decades, the technological revolution has brought about a significant change in work, speeding up its pace and increasing the information overload. The characteristics of the contemporary labor market (e.g., temporary contracts, new psychological contracts between workers and employers, and perceptions of employability) jeopardize job security, representing an inexhaustible source of occupational stress and, in chronic cases, burnout. Occupational stress results from the individual's inability to deal with the sources of stress associated with their work context. This feeling of stress can have physical, psychological, social, personal, and professional consequences for the individual (Afonso and Gomes 2012; Santos et al. 2018). In Portugal, problems related to employee stress and mental health cost organizations EUR 5.3 billion in 2022 (Ordem dos Psicólogos Portugueses 2023). This damage is caused by absenteeism, presenteeism, and loss of productivity.

Another major problem facing organizations today is the high turnover of employees, especially if they are highly specialized, which can represent high costs for the organization (Kusy and O'Leary-Driscoll 2020; Reiche 2008). This situation has led many professionals and scientists to explore the factors that have a significant impact on turnover intentions (Regts and Molleman 2013), since previous studies (Griffeth et al. 2000; Regts and Molleman

2013; van Breukelen et al. 2004) indicate that the best predictor of voluntary departure from the organization is the turnover intention. This response can be interpreted in the light of the theory of planned behavior, according to which the behavioral intentions are the causal antecedents of behavior (Ajzen 2012).

From the perspectives of Avey et al. (2009), among other factors, one of the antecedents of turnover intentions is occupational stress. Occupational stress can result in burnout, affecting job satisfaction and productivity, and leading to employees wanting to leave the organization (Salama et al. 2022). This reasoning leads us to our first objective, which is to study whether burnout is the mechanism that explains the relationship between occupational stress and turnover intentions.

Previous studies on turnover intentions have mainly focused on the relationship between this construct and organizational or individual factors such as occupational stress, burnout, job satisfaction, and job insecurity (Huang et al. 2003; Urbanaviciute et al. 2018; Wang et al. 2012). Few studies have focused on the individual motivation as an antecedent of turnover intentions (Zheng et al. 2021). However, Wang et al. (2019) studied the relationship between motivation and turnover intentions and found a significant and negative relationship between these two constructs.

This is why it was considered essential to study the relationship between motivation and turnover intentions, and whether motivation moderates the relationship between occupational stress and turnover intentions, which is our second objective. What differentiates this study from others is that the study population was employees of the Portuguese Tax and Customs Authority, working in all districts of Portugal, including the Autonomous Regions of Madeira and the Azores.

## 2. Theoretical Framework and Research Hypotheses

### 2.1. Occupational Stress and Turnover Intentions

Occupational stress can be defined as a reaction to the perception of a lack of resources to deal with a given situation (Jiang et al. 2022). Occupational stress, as an emotional state, is one of the stress issues affecting the working population and has gained enormous importance, being one of the most significant mental health problems (Bicho and Pereira 2007). This results from a group of situations and experiences at work, i.e., "the interaction between working conditions and the characteristics of the worker, in such a way that the demands placed on them exceed their ability to cope with them" (Bicho and Pereira 2007). When occupational stress becomes chronic, it can have severe consequences on both a personal and professional level, with consequences that can be both physical and psychological (Yao et al. 2019). In their work, employees face their job's demands, such as tight deadlines and high workloads, which can lead them to develop high levels of occupational stress (Imeokparia and Ediagbonya 2013; Issever et al. 2008).

Turnover intentions are understood as the employees' desires to leave their current organization and start looking for a new workplace (Benson 2006) and are the final step in the decision-making process before leaving the workplace (Bester et al. 2015). It is known that the turnover intentions are the best predictor of voluntary departure from the organization (Long et al. 2012; Moreira et al. 2022; Park and Shaw 2013), which can cause great damage to the organization especially if the departing employee is highly qualified (Kusy and O'Leary-Driscoll 2020; Reiche 2008). This relationship can be explained based on the theory of planned behavior, with the behavioral intentions being the antecedents of the actual behavior (Ajzen 2012).

As for the relationship between occupational stress and turnover intentions, employees who experience high-stress levels tend to leave the organization where they work as quickly as possible (Lee and Song 2020). There are many studies linking occupational stress with turnover intentions in professionals from various sectors such as health, hospitality, teaching, and banking (Ahn and Wang 2019; Gok et al. 2017; Gautam and Gautam 2022; Islam et al. 2019; Jabutay and Rungruang 2020; Lee and Song 2020). For example, we can cite the study by Jiang et al. (2022) on emergency medical professionals, where they concluded

that occupational stress has a positive and significant effect on turnover intentions. The results of the study by Tziner et al. (2015) also go in the same direction, i.e., a positive and significant relationship between occupational stress and turnover intentions.

This study sought to understand how occupational stress influences the intentions to leave. The following hypothesis was formulated:

**Hypothesis 1.** *Occupational stress (users, management, colleagues, overwork, career and pay, family problems, and working conditions) has a positive and significant effect on turnover intentions.*

### 2.2. Occupational Stress and Burnout

To Maslach and Leiter (2016), burnout is a psychological syndrome characterized by high levels of emotional exhaustion (such as lethargy, exhaustion, and fatigue), depersonalization (such as negative attitudes, irritability, and social withdrawal), and reduced personal fulfilment (such as decreased productivity and/or inability to cope with the situation(s)).

According to Meier's (1984) theoretical framework, burnout is viewed as a state that follows a pattern of continuous work experiences in which the subject has low expectations for the presence of rewarding stimuli, high expectations for the presence of punishment, and low expectations for their capacity to manage their efforts. With such high expectations, it is common to feel unhappy at work. According to this theoretical section, the environment and the subject interact to cause burnout rather than being caused by it alone. According to Leiter (1988), people who experience emotional tiredness respond by depersonalizing, which causes them to break the psychological commitment they uphold at work, present a poor self-assessment in terms of personal fulfilment, and result in burnout. High levels of stress relate to emotional weariness.

Burnout can be considered a psychological syndrome that occurs when employees face a stressful work environment over a long period of time and perceive their resources to cope with the demands of the job to be scarce (Bakker and Demerouti 2008; Maslach et al. 2001; Leiter and Maslach 2016). It is therefore considered to be a serious reaction to occupational stress, with resulting physical and psychological changes (Marques-Pinto et al. 2003). The consequences of burnout are varied in terms of health, safety, and well-being, as well as productivity, quality of service, and cost-effectiveness for the organization (Poghosyan et al. 2010; Carod-Artal and Vázquez-Cabrera 2013). In a study carried out by Jesus et al. (2023) on healthcare professionals, it was found that burnout has a significant effect on suicidal behavior. Occupational stress is one of the antecedents of burnout, as long-term stress leads to severe situations of exhaustion (Galanakis et al. 2020; Khalid et al. 2020). One occupational stress factor leading to severe burnout is overwork (Singh et al. 2020). Many studies on the relationship between occupational stress and burnout focus on health or education professionals. We aim to test this relationship with Portuguese Tax and Customs Authority professionals.

The following hypothesis is therefore formulated:

**Hypothesis 2.** *Occupational stress (users, management, colleagues, overwork, career and pay, family problems, and working conditions) has a significant and positive effect on burnout levels (disengagement and exhaustion).*

### 2.3. Burnout and Turnover Intentions

As organizations are aware that high employee turnover, especially if they are highly specialized, represents high costs for them (Reiche 2008), in recent years, there has been an increase in studies on the effect of burnout on increasing absenteeism rates and turnover intentions. According to Olivares-Faúndez et al. (2014), burnout is positively and significantly associated with absenteeism attitudes and behaviors. In turn, in a study of employees in the restaurant sector, Han et al. (2016) concluded that the burnout caused by customer

incivility has a positive and significant association with turnover intentions. For Rahim and Cosby (2016) as well as Kartono and Hilmiana (2018), a direct relationship was also found between burnout levels and turnover intentions. In a study of education professionals in the United States, Lee (2019) also concluded that burnout is a strong predictor of the intention to leave.

These results from previous studies led us to formulate the following hypothesis:

**Hypothesis 3.** *Burnout (disengagement and exhaustion) has a positive and significant effect on turnover intentions.*

### 2.4. Occupational Stress, Burnout, and Turnover Intentions

As previously mentioned, occupational stress has a significant and positive association with burnout, which in turn has a positive and significant association with turnover intentions. Several studies report that burnout has a mediating effect on the relationship between occupational stress and turnover intentions (Han et al. 2016; Salama et al. 2022). In a study of professionals in the education sector, Padmasundari (2019) concluded that the stress of working as an early childhood educator could be seen with the increase in burnout and intentions to leave, reducing teaching effectiveness.

We aim to test whether burnout is the mechanism that explains the relationship between occupational stress and turnover intentions. To this end, we formulated the following hypothesis:

**Hypothesis 4.** *Burnout (disengagement and exhaustion) has a mediating effect on the relationship between occupational stress (users, management, colleagues, overwork, career and pay, family problems, and working conditions) and turnover intentions.*

### 2.5. Motivation at Work and Turnover Intentions

Studies on motivation in the workplace have been the subject of great interest internationally due to the link between individual and organizational performance (Tamayo and Paschoal 2003). Maslow (2000), a pioneer of theories on motivation, highlights the importance of this issue, stating that "The healthier we are emotionally, the more important our needs for creative fulfilment at work become. At the same time, the less we tolerate the violation of our needs for such fulfilment".

Various researchers have proposed theories on motivation to understand its origin and influence on human behavior (Silva and Gomes 2009). One is the self-determination theory (Deci et al. 2017; Deci and Ryan 2008; Gagné et al. 2015). According to this theory, there are two qualitatively different forms of work motivation: autonomous and controlled. Regarding autonomous motivation, we get involved in the work because of the pleasure it gives us (intrinsic motivation) or because we identify with the task, and it has value and/or meaning for us (identified motivation) (Grohmann et al. 2013). As for controlled motivation, we get involved in the work for external reasons (extrinsic motivation) or internal reasons (introjected motivation), due to the regulation of behavior by self-esteem contingencies (Grohmann et al. 2013). Although few studies link motivation with turnover intentions, in a study on professionals in the education sector, Hussain et al. (2018) concluded that motivation has a negative and significant effect on turnover intentions and a positive effect on performance. In a more complex study of public and private sector employees, Zheng et al. (2021) found a negative and significant association between intrinsic motivation and turnover intentions. Kim (2018), in a study on public employees from local governments in South Korea, concluded that intrinsic motivation negatively and significantly affects turnover intentions. The following hypothesis was therefore formulated:

**Hypothesis 5.** *Motivation (intrinsic, identified, introjected, and extrinsic) has a negative and significant effect on turnover intentions.*

*2.6. The Moderating Effect of Motivation*

As seen above, motivation has a negative and significant effect on turnover intentions. Furthermore, in a study carried out by Özbağ et al. (2014), they concluded that motivation moderates the relationship between burnout and turnover intentions. However, one of the aims of this study is to determine if motivation has a moderating effect on the relationship between occupational stress and turnover intentions. The following hypothesis was therefore formulated:

**Hypothesis 6.** *Motivation (intrinsic, identified, introjected, and extrinsic) moderates the relationship between occupational stress (users, management, colleagues, overwork, career and pay, family problems, and working conditions) and turnover intentions, with the relationship expected to be weaker for employees with high levels of motivation than for employees with low levels of motivation.*

To integrate the hypotheses formulated in this study, a theoretical model was developed that synthesized all the relationships (Figure 1).

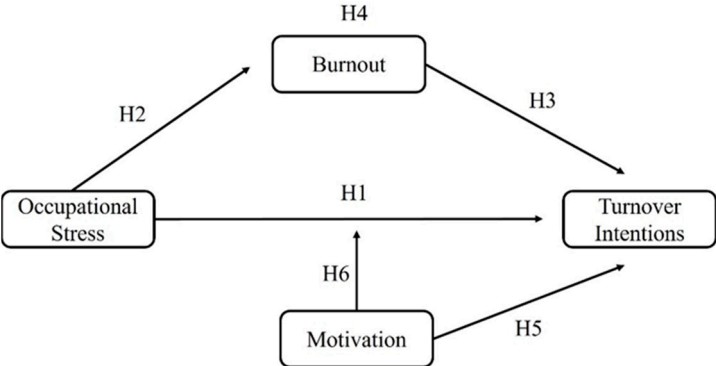

**Figure 1.** Research model.

## 3. Materials and Methods

### 3.1. Data Collection Procedure

A total of 603 subjects voluntarily participated in this study, all employees of the Portuguese Tax and Customs Authority, which is part of the Ministry of Finance and belongs to the Direct State Administration. The data collection process was nonprobabilistic, intentional, and followed snowball sampling (Trochim 2000).

The questionnaire was posted online on the Google Forms platform, and its link was sent to employees of the Portuguese Customs Authority via the Trade Union Association of Tax and Customs Inspection Professionals (APIT). A member of the APIT (coauthor of this article) sent the link to the questionnaire to all Portuguese Tax and Customs Authority employees working in mainland Portugal and the Autonomous Regions of Madeira and the Azores. The questionnaire contained all the information about the purpose of the study, as well as guaranteed the confidentiality of the answers provided. Participants were also informed that their answers would never be known since the data would be processed considering all the answers provided. At the beginning of the questionnaire, after reading the informed consent form, participants were asked if they agreed to answer the questionnaire. If participants chose the "no" option, they were directed to the end of the questionnaire, i.e., to the acknowledgements page.

The questionnaire included ten sociodemographic questions (age, gender, academic qualifications, length of service in the tax and customs authority, length of service in the civil service, district of residence, service to which they belonged, type of employment, staff group in the tax administration, method of admission, and whether they were unionized) and four scales (occupational stress, burnout, motivation, and turnover intentions). The data were collected between April and June 2023.

### 3.2. Participants

The study sample consisted of 603 participants between the ages of 32 and 69 (M = 52.46; SD = 6.86), with 320 (53.1%) female participants and 283 (46.9%) male participants. In terms of educational qualifications, 155 (25.7%) have a 12th-grade degree or less, 396 (65.7%) have a bachelor's degree, and 52 (8.6%) have a master's degree or higher. Seniority in the customs authority varies between 1 and 46 years (M = 23.49; SD = 10.66), and seniority in the civil service varies between 5 and 47 years (M = 27.09; SD = 8.29). Concerning the district in which they live, the highest percentage of participants work in the districts of Lisbon or Porto, although the employees work in all of Portugal's districts (Table 1).

**Table 1.** Distribution of employees by district in Portugal.

| District | Frequency | Percentage |
|---|---|---|
| Faro | 20 | 3.3 |
| Beja | 5 | 0.8 |
| Évora | 2 | 0.3 |
| Setúbal | 41 | 6.8 |
| Portalegre | 9 | 1.5 |
| Santarém | 28 | 4.6 |
| Lisboa | 159 | 26.4 |
| Castelo Branco | 12 | 2.0 |
| Leiria | 55 | 9.1 |
| Coimbra | 39 | 6.5 |
| Guarda | 10 | 1.7 |
| Viseu | 18 | 3.0 |
| Aveiro | 45 | 7.5 |
| Porto | 86 | 14.3 |
| Bragança | 5 | 0.8 |
| Vila Real | 10 | 1.7 |
| Braga | 32 | 5.3 |
| Viana do Castelo | 7 | 1.2 |
| Funchal | 6 | 1.0 |
| Ponta Delgada | 5 | 0.8 |
| Horta | 4 | 0.7 |
| Angra do Heroísmo | 5 | 0.8 |

Among these participants, 71 (11.8%) work in central services, 212 (35.2%) in finance directorates, 70 (11.6) in customs, 233 (38.6%) in finance services, and 17 (2.8%) in customs offices. As for the method of employment, 264 (43.8%) are appointed, 16 (2.7%) are on service commission, and 323 (53.6%) are on public service contracts. Among these participants, 14 (2.3%) are managers, 67 (11.1%) are in tax and customs management, 262 (43.4%) are in tax and customs inspection/management, 133 (22.1%) belong to subsistence careers, 88 (14.6%) belong to IT staff, and 39 (6.5%) to general regime staff. As for how they were hired, 536 (88.9%) were hired through a competitive procedure, 38 (6.3%) through internal mobility, 2 (0.3%) returned from unpaid leave, 12 (2%) are on service commission and 15 (205%) are in another situation. Regarding whether they are union members, 487 (80.8%) said they are union members, and 116 (19.2%) said they are not union members.

### 3.3. Data Analysis Procedure

The data were imported into SPSS Statistics 29.0 software (IBM Corp., Armonk, NY, USA). The first step was to test the metric qualities of the instruments used in this study. To test the validity of the instruments measuring occupational stress, motivation, and burnout, confirmatory factor analyses were carried out using AMOS Graphics 29.0 software (IBM Corp., Armonk, NY, USA). The procedure followed a "model generation" logic (Jöreskog and Sörbom 1993). Six fit indices were merged following the published recommendations (Hu and Bentler 1999), considering in the analysis of their adjustment the results obtained for the chi-square test ($\chi^2/df \leq 5$); for the Tucker Lewis index (TLI) > 0.90; for the goodness-

of-fit index (GFI) > 0.90; for the comparative fit index (CFI) > 0.90; for the root mean square error of approximation (RMSEA) ≤ 0.08; and for the root mean square residual (RMSR), where a smaller value corresponds to a better adjustment. We then tested the construct reliability for each scale's dimensions, which should be greater than 0.70. Finally, the convergent validity was tested by calculating the average variance extracted (AVE), which should be greater than 0.50 (Fornell and Larcker 1981). However, when Cronbach's alpha value is above 0.70, AVE values greater than 0.40 are acceptable, indicating good convergent validity (Hair et al. 2011).

The discriminant validity was tested by comparing the square root of the AVE values with the correlation values between factors. The square root values of the AVE should be higher than the correlation value between the factors whose discriminant validity is to be analyzed.

For the instrument measuring turnover intentions, as it is made up of only three factors, its validity was tested by carrying out an exploratory factor analysis. The KMO value was calculated, which should be greater than 0.70 (Sharma 1996). We also calculated the average variance extracted, which should be greater than 50%. As for the factor weights of each item, all items with factor weights greater than 0.50 were considered.

The internal consistency was tested for each of the dimensions that make up the instrument by calculating Cronbach's alpha, which must be greater than 0.70 (Bryman and Cramer 2003).

Concerning the sensitivity of the items, the median, minimum, maximum, asymmetry, and kurtosis were calculated. Items should not have the median leaning against one of the extremes, and they should have responses at all points, and their absolute values of skewness and kurtosis should be below 3 and 7, respectively (Kline 2011).

Descriptive statistics were carried out on the variables under study to see if the answers given by the participants differed significantly from the central point of the respective scale. This was done using the one-sample Student's *t*-test. The association between the sociodemographic variables and the variables under study was also tested using Student's *t*-test for independent samples, one-way ANOVA, and Pearson's correlation.

Hypotheses 1, 2, 3, 4, and 5 were tested using simple and multiple linear regressions. Hypothesis 6, which assumed a moderating effect, was tested using MACRO Process 4.0, developed by Hayes (2013).

### 3.4. Instruments

To measure occupational stress, we used the Occupational Stress Instrument—General Version (QSO-VG) by Gomes (2010), developed based on studies in different professional areas, giving it credibility for studying occupational stress. This instrument identifies and assesses potential sources of stress (i.e., stressors) in the course of work. It consists of 24 items relating to sources of stress. These items are divided into seven subscales with a 5-point Likert score (0 = no stress to 4 = much stress). The dimensions assessed are as follows: relationship with users (items 2, 8, 13, and 21); relationship with managers (items 12, 20, and 24); relationship with colleagues (items 4, 17, and 22); overwork (items 5, 10, 11, and 16); career and pay (items 1, 6, 15, and 19); family problems (items 3, 14, and 23); and working conditions (items 7, 9, and 18). A confirmatory factor analysis was carried out on the seven factors. The fit indices obtained were adequate ($\chi^2/gl$ = 2.90; GFI = 0.93; CFI = 0.97; TLI = 0.96; RMSEA = 0.056; and SRMR = 0.045). The construct reliability of the dimensions varied between 0.85 (relationship with management) and 0.96 (family problems). Regarding convergent validity, the AVE values varied between 0.65 (relationship with managers) and 0.88 (family problems). As for divergent validity, all the AVE's square root values were higher than the correlations between the factors for which discriminant validity was to be tested.

To measure turnover intentions, we used the instrument developed by Bozeman and Perrewé (2001), consisting of 3 items, which are anchored on a five-point Likert scale (from 1 "Strongly Disagree" to 5 "Strongly Agree"). The validity of this instrument was tested

using an exploratory factor analysis. A KMO of 0.74 was obtained, which can be considered reasonable (Sharma 1996), and Bartlett's test of sphericity was significant at $p < 0.001$, which indicates that the sample comes from a multivariate population (Pestana and Gageiro 2003). This instrument is made up of one factor, which explains 85.32% of its total variability. As for internal consistency, a Cronbach's alpha of 0.91 was obtained.

The Oldenburg Burnout Inventory, developed by Demerouti and Nachreiner (1998) and adapted for the Portuguese population by Sinval et al. (2019), measured burnout. Its aim is to assess the exhaustion and depersonalization dimensions (Halbesleben and Demerouti 2005), in which the exhaustion subscale represents the feeling of emptiness, excessive workload, physical, cognitive, and emotional exhaustion (Demerouti et al. 2003). The depersonalization subscale reflects the disengagement from the professional environment and the attitudes towards work (Bakker and Demerouti (2008). This instrument consists of 16 items, which are anchored on a five-point Likert scale (from 1 "Strongly Disagree" to 5 "Strongly Agree"). These 16 items are divided into two dimensions: disengagement (items 1, 3, 6, 7, 9, 11, 13, and 15) and exhaustion (items 2, 4, 5, 8, 10, 12, 14, and 16). The validity of this instrument was tested by carrying out a two-factor confirmatory factor analysis. The fit indices obtained were adequate ($\chi^2/\text{gl} = 3.81$; GFI = 0.95; CFI = 0.95; TLI = 0.93; RMSEA = 0.068; and SRMR = 0.045). The construct reliability for disengagement was 0.85, and for exhaustion, 0.91. As for internal consistency, disengagement had a Cronbach's alpha of 0.84 and exhaustion of 0.88. Concerning convergent validity, the AVE values were 0.43 for disengagement and 0.56 for exhaustion. Although disengagement had a low AVE value, according to Hair et al. (2011), when Cronbach's alpha value is above 0.70, values above 0.40 can be accepted. As for divergent validity, all the AVE's square root values were higher than the correlation between the two factors.

Motivation was measured using the instrument developed by Gagné et al. (2010), consisting of 12 items anchored on a five-point Likert scale (from 1 "Strongly Disagree" to 5 "Strongly Agree"). This instrument is made up of 4 dimensions: intrinsic motivation (items 1, 2, and 3); identified motivation (items 4, 5, and 6); introjected motivation (items 7, 8, and 9); and extrinsic motivation (items 10, 11, and 12). To test the validity of this instrument, a 4-factor confirmatory factor analysis was carried out. The fit indices obtained were adequate ($\chi^2/\text{gl} = 2.76$; GFI = 0.97; CFI = 0.98; TLI = 0.97; RMSEA = 0.054; and SRMR = 0.033). As the extrinsic motivation dimension had very low construct reliability (0.61), very low internal consistency (0.59), and a low AVE value (0.43), we decided to remove this dimension. A new confirmatory factor analysis was carried out. The fit indices were adequate ($\chi^2/\text{gl} = 2.42$; GFI = 0.98; CFI = 0.99; TLI = 0.98; RMSEA = 0.049; and SRMR = 0.030). The construct reliability varied between 0.79 (introjected motivation) and 0.84 (intrinsic motivation). As for consistency, it varied between 0.78 (introjected motivation) and 0.86 (intrinsic motivation). Regarding convergent validity, the AVE values varied between 0.57 (introjected motivation) and 0.64 (intrinsic motivation). As for divergent validity, all the AVE's square root values were higher than the factor correlation.

## 4. Results

### 4.1. Descriptive Statistics of the Variables under Study

To understand the position of the answers given by the participants, descriptive statistics were carried out on the variables under study.

The results show that the participants in this study reported levels of stress, with all the dimensions having an average significantly higher than the central point of this scale (2), except for the "stress caused by colleagues" dimension (Table 2). It was the employees working in the customs officers who showed the highest levels of stress in terms of stress about career and pay, family problems, and working conditions (Figure 2). The employees working in finance had higher levels of stress caused by users and overwork (Figure 2). Concerning the stress caused by colleagues and managers, the employees working in customs reported the highest levels (Figure 2).

**Table 2.** Descriptive statistics of the variables under study.

| Scale | Dimension | t | *p* | Mean | SD |
|---|---|---|---|---|---|
| Occupational stress | Users | 11.88 | <0.001 | 2.46 | 0.96 |
| | Managers | 7.17 | <0.001 | 2.32 | 1.10 |
| | Colleagues | −3.84 | <0.001 | 1.83 | 1.08 |
| | Overwork | 21.42 | <0.001 | 2.77 | 0.88 |
| | Career and pay | 22.85 | <0.001 | 2.83 | 0.90 |
| | Family problems | 9.87 | <0.001 | 2.43 | 1.07 |
| | Working conditions | 18.83 | <0.001 | 2.74 | 0.97 |
| Turnover intentions | | 11.32 | <0.001 | 3.51 | 1.11 |
| Burnout | Disengagement | 15.38 | <0.001 | 3.46 | 0.73 |
| | Exhaustion | 21.75 | <0.001 | 3.64 | 0.72 |
| Motivation | Intrinsic | −8.49 | <0.001 | 2.69 | 0.89 |
| | Identified | 10.91 | <0.001 | 3.67 | 1.51 |
| | Introjected | −15.72 | <0.001 | 2.44 | 0.88 |

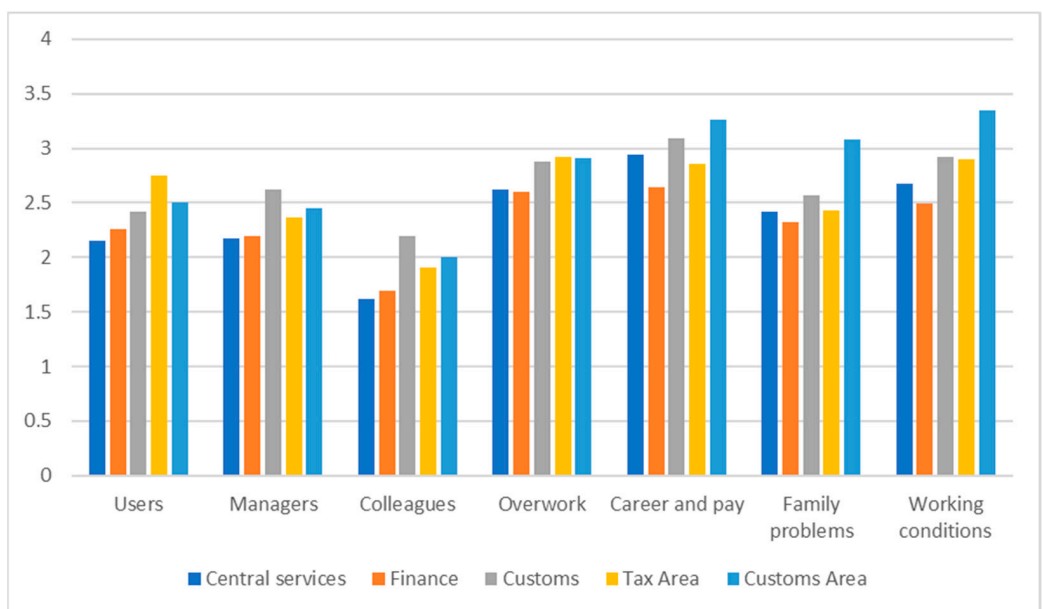

**Figure 2.** Distribution of occupational stress according to the department where the employee works.

The participants also showed high turnover intentions as well as high levels of burnout, significantly higher than the scale's central point (3) (Table 2). It is the participants working in financial services who have the highest turnover intentions (Figure 3). Regarding burnout levels, participants working in the tax area reported higher levels of disengagement, and those working in the customs area reported higher levels of exhaustion (Figure 3).

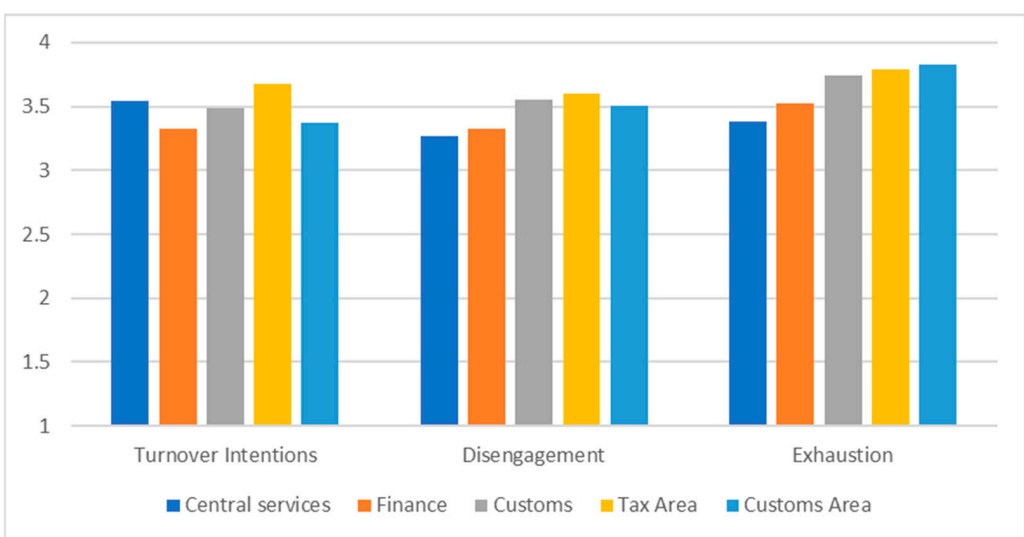

**Figure 3.** Distribution of turnover intentions and burnout according to the department where the employee works.

As for motivation levels, the intrinsic and introjected motivation were significantly below the central point of the scale (3), while the identified motivation was significantly above the central point (Table 2). The participants working in central services reported the highest levels of intrinsic motivation (Figure 4). As for identified motivation, the employees who reported the highest levels were those working in finance, and those who reported the lowest were those working in customs (Figure 4). Concerning introjected motivation, the participants working in customs reported the highest levels (Figure 4).

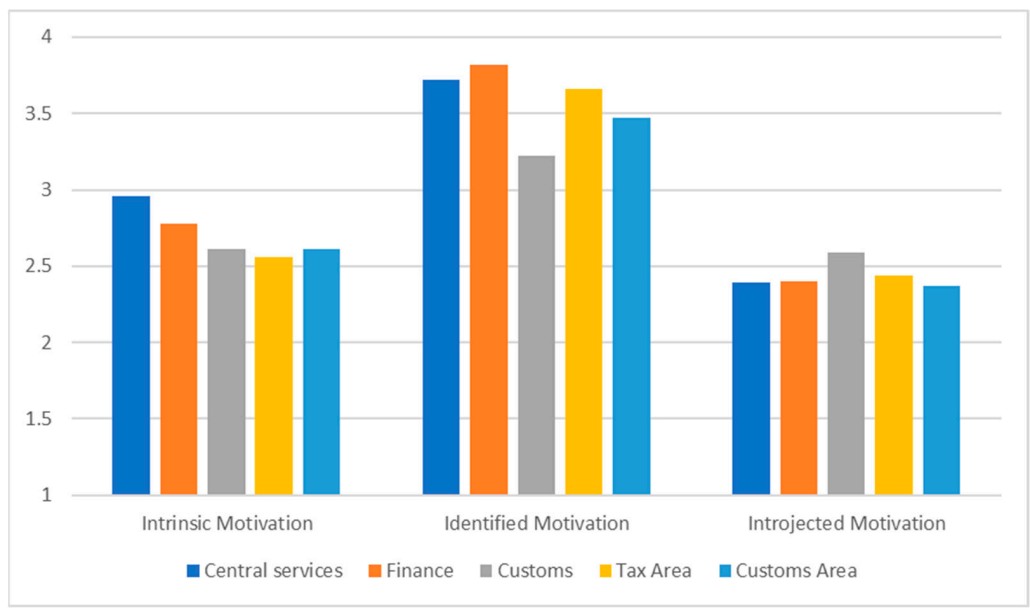

**Figure 4.** Distribution of motivation according to the department where the employee works.

### 4.2. Association between the Variables under Study

The association between the variables under study was tested using Pearson's correlation.

The results show that intrinsic and identified motivation are negatively and significantly associated with turnover intentions, disengagement, exhaustion, and all the dimensions of occupational stress (Table 3). Introjected motivation is negatively and significantly associated with turnover intentions, disengagement, and exhaustion, and positively and significantly associated with family problems (Table 3). Turnover intentions were positively

and significantly associated with disengagement, exhaustion, and all the dimensions of occupational stress (Table 3). Disengagement and exhaustion are positively and significantly associated with all dimensions of occupational stress (Table 3). The age was only negatively and significantly associated with intrinsic motivation, introjected motivation, and stress caused by career and pay (Table 3).

**Table 3.** Pearson's correlation results.

| | 1 | 2 | 3 | 4 | 5 | 6 | 7 | 8 | 9 | 10 | 11 | 12 | 13 | 14 |
|---|---|---|---|---|---|---|---|---|---|---|---|---|---|---|
| 1.　Intrinsic | - | | | | | | | | | | | | | |
| 2.　Identified | 0.38 ** | - | | | | | | | | | | | | |
| 3.　Introjected | 0.29 ** | 0.12 ** | - | | | | | | | | | | | |
| 4.　Turnover int. | −0.60 ** | −0.43 ** | −0.17 ** | - | | | | | | | | | | |
| 5.　Disengagement | −0.74 ** | −0.44 ** | −0.27 ** | 0.65 ** | - | | | | | | | | | |
| 6.　Exhaustion | −0.58 ** | −0.41 ** | −0.09 * | 0.54 ** | 0.63 ** | - | | | | | | | | |
| 7.　Users | −0.32 ** | −0.30 ** | −0.02 | 0.36 ** | 0.41 ** | 0.50 ** | - | | | | | | | |
| 8.　Managers | −0.37 ** | −0.42 ** | −0.01 | 0.41 ** | 0.48 ** | 0.50 ** | 0.48 ** | - | | | | | | |
| 9.　Colleagues | −0.28 ** | −0.30 ** | 0.00 | 0.33 ** | 0.33 ** | 0.39 ** | 0.41 ** | 0.63 ** | - | | | | | |
| 10.　Overwork | −0.34 ** | −0.37 ** | 0.03 | 0.42 ** | 0.40 ** | 0.65 ** | 0.60 ** | 0.52 ** | 0.44 ** | - | | | | |
| 11.　Career and pay | −0.22 ** | −0.51 ** | 0.06 | 0.35 ** | 0.33 ** | 0.35 ** | 0.39 ** | 0.40 ** | 0.29 ** | 0.43 ** | - | | | |
| 12.　Family probl. | −0.22 ** | −0.35 ** | 0.10 * | 0.32 ** | 0.29 ** | 0.56 ** | 0.47 ** | 0.46 ** | 0.41 ** | 0.68 ** | 0.42 ** | - | | |
| 13.　Working cond. | −0.22 ** | −0.43 ** | −0.01 | 0.34 ** | 0.36 ** | 0.43 ** | 0.51 ** | 0.50 ** | 0.41 ** | 0.59 ** | 0.50 ** | 0.50 ** | - | |
| 14.　Age | −0.10 * | 0.03 | 0.15 ** | 0.02 | 0.04 | 0.06 | −0.03 | 0.02 | 0.04 | 0.02 | −0.16 ** | 0.01 | −0.03 | - |

Note. * $p < 0.05$; ** $p < 0.01$.

### 4.3. Hypothesis

Initially, we considered age as a control variable. However, as it is not significantly correlated with any dependent variables (exhaustion, disengagement, and turnover intentions), we did not do so.

To test Hypothesis 1, a multiple linear regression was carried out.

The results show that only stress caused by the manager ($\beta = 0.17$; $p = 0.001$), overwork ($\beta = 0.21$; $p < 0.001$), and the career and remuneration ($\beta = 0.15$; $p < 0.001$) has a positive and significant effect on turnover intentions (Table 4). The model explains 24% of the variability in turnover intentions (Table 4). The results also indicate that the model is statistically significant (F (7, 595) = 28.30; $p < 0.001$) (Table 4). Hypothesis 1 was partially supported.

**Table 4.** Effect of occupational stress on turnover intentions.

| Independent Variable | Dependent Variable | F | *p* | R2a | β | *p* |
|---|---|---|---|---|---|---|
| Users | | | | | 0.08 | 0.099 |
| Managers | | | | | 0.17 ** | 0.001 |
| Colleagues | | | | | 0.06 | 0.184 |
| Overwork | Turnover intentions | 28.30 *** | <0.001 | 0.24 | 0.21 *** | <0.001 |
| Career and pay | | | | | 0.15 *** | <0.001 |
| Family problems | | | | | −0.04 | 0.481 |
| Working conditions | | | | | 0.01 | 0.861 |

Note. ** $p < 0.01$; *** $p < 0.001$.

Hypothesis 2 was tested by performing two multiple linear regressions.

The stress caused by users ($\beta = 0.16$; $p < 0.001$), managers ($\beta = 0.32$; $p < 0.001$), overwork ($\beta = 0.14$; $p = 0.012$), and career and pay ($\beta = 0.10$; $p = 0.012$) has a positive and significant effect on disengagement (Table 5). The model explains 29% of the variability in disengagement and is statistically significant (F (7, 595) = 35.40; $p < 0.001$) (Table 5).

**Table 5.** Effect of occupational stress on burnout.

| Independent Variable | Dependent Variable | F | *p* | R2a | β | *p* |
|---|---|---|---|---|---|---|
| Users | | | | | 0.16 *** | <0.001 |
| Managers | | | | | 0.32 *** | <0.001 |
| Colleagues | | | | | 0.01 | 0.970 |
| Overwork | Disengagement | 35.40 *** | <0.001 | 0.29 | 0.14 * | 0.012 |
| Career and pay | | | | | 0.10 * | 0.012 |
| Family problems | | | | | −0.08 | 0.091 |
| Working conditions | | | | | 0.03 | 0.568 |
| Users | | | | | 0.11 ** | 0.004 |
| Managers | | | | | 0.17 *** | <0.001 |
| Colleagues | | | | | 0.00 | 0.966 |
| Overwork | Exhaustion | 79.76 *** | <0.001 | 0.48 | 0.38 *** | <0.001 |
| Career and pay | | | | | 0.02 | 0.669 |
| Family problems | | | | | 0.19 *** | <0.001 |
| Working conditions | | | | | −0.04 | 0.273 |

Note. * $p < 0.05$; ** $p < 0.01$; *** $p < 0.001$.

The stress caused by users (β = 0.11; $p$ = 0.004), managers (β = 0.17; $p < 0.001$), overwork (β = 0.38; $p < 0.001$), and family problems (β = 0.19; $p < 0.001$) has a positive and significant effect on exhaustion (Table 5). The model explains 48% of the variability in exhaustion and is statistically significant (F (7, 595) = 79.76; $p < 0.001$) (Table 5). Hypothesis 2 was partially supported.

Hypothesis 3 was tested by performing a multiple linear regression.

Both disengagement (β = 0.52; $p < 0.001$) and exhaustion (β = 0.21; $p < 0.001$) have a positive and significant effect on turnover intentions (Table 5). The model explains 45% of the variability in turnover intentions and is statistically significant (F (2, 600) = 249.87; $p < 0.001$) (Table 6). Hypothesis 3 was supported.

**Table 6.** Effect of burnout on turnover intentions.

| Independent Variable | Dependent Variable | F | *p* | R2a | β | *p* |
|---|---|---|---|---|---|---|
| Disengagement | Turnover intentions | 249.87 *** | <0.001 | 0.45 | 0.52 *** | <0.001 |
| Exhaustion | | | | | 0.21 *** | <0.001 |

Note. *** $p < 0.001$.

Hypothesis 4, which presupposes a mediating effect, followed the assumptions of Baron and Kenny (1986). These assumptions were verified in Hypotheses 1, 2, and 3. Only the mediating effects between the variables that met the three assumptions were tested. Two multiple linear regressions were carried out. In each multiple linear regression, the predictor variables were introduced as the independent variables in the first step and the mediating variables in the second step.

A total mediation effect of disengagement was proven in the relationship between the stress caused by managers and turnover intentions ($β_1 = 0.22$; $β_2 = 0.02$) (Table 7). After the mediating variable was introduced into the regression equation, the stress caused by managers no longer had a significant effect on turnover intentions. Sobel's test confirmed the total mediation effect (Z = 5.81; $p < 0.001$).

**Table 7.** The mediating effect of disengagement on the relationship between occupational stress and turnover intentions.

| Independent Variable | Step 1 β | Step 2 β |
|---|---|---|
| Managers | 0.22 *** | 0.02 |
| Overwork | 0.24 *** | 0.14 *** |
| Career and pay | 0.16 *** | 0.10 ** |
| Disengagement | | 0.55 *** |
| F | 64.07 *** | 129.79 *** |
| R2a | 0.24 | 0.46 |
| ΔR2a | | 0.22 *** |

Note. ** $p < 0.01$; *** $p < 0.001$.

As for the mediating effect of disengagement on the relationship between the stress caused by overwork and turnover intentions ($\beta_1 = 0.24$; $\beta_2 = 0.14$), as well as on the relationship between the stress caused by career and pay and turnover intentions ($\beta_1 = 0.16$; $\beta_2 = 0.10$), there were two partial mediation effects; when the mediating variable was introduced into the regression equation, both overwork stress and career and pay stress continued to have a significant effect on the dependent variable, but the effect decreased in intensity (Table 7). Sobel's test confirmed the partial mediation effect for overwork stress ($Z = 2.49$; $p = 0.013$) and work overload, as well as career and pay stress ($Z = 2.46$; $p = 0.013$). The increase in variability proved to be significant ($\Delta R^2_a = 0.22$; $p < 0.001$). Both model 1 (F (3, 599) = 64.07; $p < 0.001$) and model 2 (F (4, 598) = 129.79; $p < 0.001$) were statistically significant (Table 7).

A total mediation effect of exhaustion was proven in the relationship between overwork stress and turnover intentions ($\beta_1 = 0.29$; $\beta_2 = 0.06$), since after the mediating variable was introduced into the regression equation, overwork stress no longer had a significant effect on turnover intentions (Table 8). Sobel's test confirmed the total mediation effect ($Z = 4.49$; $p < 0.001$).

**Table 8.** The mediating effect of exhaustion on the relationship between occupational stress and turnover intentions.

| Independent Variable | Step 1 β | Step 2 β |
|---|---|---|
| Managers | 0.26 *** | 0.16 *** |
| Overwork | 0.29 *** | 0.06 |
| Exhaustion | | 0.42 *** |
| F | 86.07 *** | 93.40 *** |
| R2a | 0.22 | 0.32 |
| ΔR2a | | 0.10 *** |

Note. *** $p < 0.001$.

As for the mediating effect of exhaustion on the relationship between the stress caused by the manager and turnover intentions ($\beta_1 = 0.24$; $\beta_2 = 0.14$), a partial mediation effect was confirmed, since when the mediating variable was introduced into the regression equation, the stress caused by the manager continued to have a significant effect on the dependent variable, but the effect decreased in intensity (Table 8). Sobel's test confirmed the partial mediation effect ($Z = 3.24$; $p = 0.001$). The increase in variability proved to be significant ($\Delta R^2_a = 0.10$; $p < 0.001$). Both model 1 (F (2, 600) = 86.07; $p < 0.001$) and model 2 (F (3, 599) = 93.40; $p < 0.001$) were statistically significant (Table 8).

Hypothesis 4 was partially supported.

Hypothesis 5 was tested by performing a multiple linear regression.

The results show that intrinsic motivation ($\beta = -0.51$; $p < 0.001$) and identified motivation ($\beta = -0.24$; $p < 0.001$) have a negative and significant effect on turnover intentions

(Table 9). The model explains 40% of the variability in turnover intentions and is statistically significant (F (3, 599) = 133.76; $p < 0.001$). Hypothesis 5 was partially supported.

**Table 9.** Effect of motivation on turnover intentions.

| Independent Variable | Dependent Variable | F | *p* | R2a | β | *p* |
|---|---|---|---|---|---|---|
| Intrinsic motivation | | | | | −0.51 *** | <0.001 |
| Identified motivation | Turnover intentions | 133.76 *** | <0.001 | 0.40 | −0.24 *** | <0.001 |
| Introjected motivation | | | | | 0.01 | 0.886 |

Note. *** $p < 0.001$.

To test hypothesis 6, as it presupposes a moderation effect, the Macro Process developed by Hayes (2013) was used. Only the moderating effect of intrinsic motivation and identified motivation on the relationship between occupational stress and turnover intentions was tested since identified motivation has no significant effect on turnover intentions.

The results showed that intrinsic motivation has a moderating effect on the relationship between occupational stress (with managers, colleagues, career and pay, and family problems) and turnover intentions (Table 10).

**Table 10.** Moderating effect of Intrinsic Motivation on the relationship between Occupational Stress and Turnover Intentions.

| Variables | B | SE | t | *p* | 95% CI |
|---|---|---|---|---|---|
| | Users → turnover intentions (R$^2$ = 0.39; $p < 0.001$) | | | | |
| Constant | 3.53 *** | 0.04 | 95.50 *** | <0.001 | [3.46, 3.60] |
| Users | 0.21 *** | 0.04 | 5.39 *** | <0.001 | [0.13, 0.29] |
| Intrinsic motivation | −0.67 *** | 0.04 | −15.93 *** | <0.001 | [−0.76, −0.59] |
| U*IntM | −0.06 | 0.04 | −1.64 | 0.101 | [−0.13, 0.14] |
| | Managers → turnover intentions (R$^2$ = 0.40; $p < 0.001$) | | | | |
| Constant | 3.54 *** | 0.04 | 94.73 *** | <0.001 | [3.46, 3.60] |
| Managers | 0.21 *** | 0.03 | 6.09 *** | <0.001 | [0.14, 0.27] |
| Intrinsic motivation | −0.65 *** | 0.04 | −15.22 *** | <0.001 | [−0.74, −0.57] |
| M*IntM | −0.07 * | 0.04 | −2.05 * | 0.041 | [−0.01, −0.14] |
| | Colleagues → turnover intentions (R$^2$ = 0.39; $p < 0.001$) | | | | |
| Constant | 3.54 *** | 0.04 | 96.64 *** | <0.001 | [3.46, 3.61] |
| Colleagues | 0.19 *** | 0.03 | 5.53 *** | <0.001 | [0.12, 0.25] |
| Intrinsic motivation | −0.68 *** | 0.04 | −16.52 *** | <0.001 | [−0.77, −0.60] |
| C*IntM | −0.10 ** | 0.04 | −2.64 ** | 0.009 | [−0.02, −0.17] |
| | Overwork → turnover intentions (R$^2$ = 0.41; $p < 0.001$) | | | | |
| Constant | 3.52 *** | 0.04 | 96.11 *** | <0.001 | [3.45, 3.58] |
| Overwork | 0.31 *** | 0.04 | 7.22 *** | <0.001 | [0.22, 0.39] |
| Intrinsic motivation | −0.64 *** | 0.04 | −15.30 *** | <0.001 | [−0.72, −0.56] |
| O*IntM | −0.03 | 0.04 | −0.67 | 0.503 | [−0.06, 0.12] |
| | Career and pay → turnover intentions (R$^2$ = 0.41; $p < 0.001$) | | | | |
| Constant | 3.53 *** | 0.04 | 99.16 *** | <0.001 | [3.46, 3.60] |
| Career and pay | 0.28 *** | 0.04 | 6.92 *** | <0.001 | [0.20, 0.35] |
| Intrinsic motivation | −0.69 *** | 0.04 | −17.05 *** | <0.001 | [−0.77, −0.61] |
| CP*IntM | −0.10 * | 0.04 | −2.31 * | 0.022 | [−0.02, −0.19] |
| | Family problems → turnover intentions (R$^2$ = 0.39; $p < 0.001$) | | | | |
| Constant | 3.53 *** | 0.04 | 97.88 *** | <0.001 | [3.46, 3.60] |
| Family problems | 0.20 *** | 0.03 | 5.83 *** | <0.001 | [0.13, 0.26] |
| Intrinsic motivation | −0.70 *** | 0.04 | −17.04 *** | <0.001 | [−0.78, −0.62] |
| FP*IntM | −0.08 * | 0.04 | −2.10 * | 0.036 | [−0.01, −0.15] |
| | Work conditions→ turnover intentions (R$^2$ = 0.40; $p < 0.001$) | | | | |
| Constant | 3.52 *** | 0.04 | 98.23 *** | <0.001 | [3.45, 3.59] |
| Work conditions | 0.24 *** | 0.03 | 6.50 *** | <0.001 | [0.17, 0.32] |
| Intrinsic motivation | −0.69 *** | 0.04 | −16.86 *** | <0.001 | [−0.77, −0.61] |
| WC*IntM | −0.05 | 0.04 | −1.25 | 0.213 | [−0.03, 0.13] |

Note. * $p < 0.05$; ** $p < 0.01$; *** $p < 0.001$.

For participants with low levels of intrinsic motivation, when compared to participants with high intrinsic motivation, occupational stress (caused by managers, colleagues, career

and pay, and family problems) becomes relevant to boosting their turnover intentions (Figure 5).

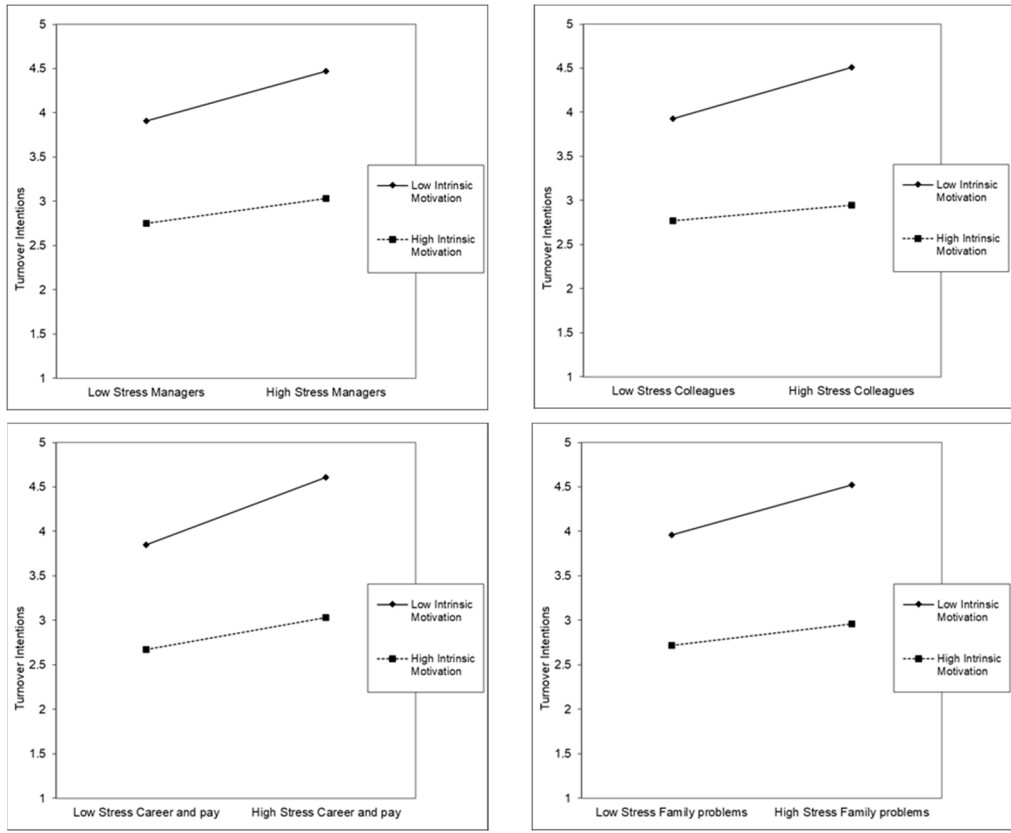

**Figure 5.** Interaction effects of occupational stress $\times$ intrinsic motivation.

There was no moderating effect of identified motivation on the relationship between occupational stress and turnover intentions. Hypothesis 6 was partially supported (Table A1). Finally, Table 11 summarizes the results obtained in this study.

**Table 11.** Summary of hypotheses results.

| | | |
|---|---|---|
| H1 | Occupational stress (users, management, colleagues, overwork, career and pay, family problems, and working conditions) has a positive and significant effect on turnover intentions. | Partially supported |
| H2 | Occupational stress (users, management, colleagues, overwork, career and pay, family problems, and working conditions) has a significant and positive effect on burnout levels (disengagement and exhaustion). | Partially supported |
| H3 | Burnout (disengagement and exhaustion) has a positive and significant effect on turnover intentions. | Supported |
| H4 | Burnout (disengagement and exhaustion) has a mediating effect on the relationship be-tween occupational stress (users, management, colleagues, overwork, career and pay, family problems, and working conditions) and turnover intentions. | Partially supported |
| H5 | Motivation (intrinsic, identified, introjected, and extrinsic) has a negative and significant effect on turnover intentions. | Partially supported |
| H6 | Motivation (intrinsic, identified, introjected, and extrinsic) moderates the relationship between occupational stress (users, management, colleagues, overwork, career and pay, family problems, and working conditions) and turnover intentions, with the relationship expected to be weaker for employees with high levels of motivation than for employees with low levels of motivation. | Partially supported |

## 5. Discussion

The main aim of this study was to investigate the relationship between occupational stress and turnover intentions and whether this relationship was mediated by burnout and moderated by motivation.

Hypothesis 1 was partially supported since among the dimensions of occupational stress, only the stress caused by managers, overwork, and career and pay has a positive and significant effect on turnover intentions. These results align with the literature that occupational stress has a positive and significant effect on turnover intentions (Jiang et al. 2022). The dimension of occupational stress that has the strongest effect on turnover intentions is work overload. This result aligns with what Singh et al. (2020) say, that work overload is one of the most critical factors in occupational stress levels (Imeokparia and Ediagbonya 2013).

Hypothesis 2 was also partially supported. The stress caused by users, managers, work overload, and career and remuneration has a positive and significant effect on disengagement. It should be noted that the dimension of occupational stress that has the strongest effect on disengagement is the stress caused by managers. The stress caused by users, managers, work overload, and family problems has a positive and significant effect on exhaustion. The dimension of occupational stress that has the strongest effect on burnout is work overload. These results are in line with the literature since burnout occurs when employees face a stressful work environment over a long period, and there are few resources to cope with the demands of the job (Bakker and Demerouti 2008; Galanakis et al. 2020; Khalid et al. 2020; Maslach and Leiter 2016; Maslach et al. 2001). Burnout is a severe reaction to occupational stress (Marques-Pinto et al. 2003), and one of the occupational stress factors that leads to more severe burnout situations (especially exhaustion) is work overload (Singh et al. 2020).

As expected, Hypothesis 3 was confirmed. Both disengagement and burnout have a positive and significant effect on turnover intentions. It should be noted that disengagement is the dimension that has the most substantial effect on turnover intentions, which is natural because if the employee is in a situation of disengagement, it is natural that they will be more willing to leave the organization where they are working. These results are in line with what the literature tells us. According to several authors, burnout has a positive and significant effect on turnover intentions (Rahim and Cosby 2016; Kartono and Hilmiana 2018; Lee 2019).

As for the mediating effect of burnout on the relationship between occupational stress and turnover intentions, it was found that disengagement has a total mediating effect on the relationship between the stress caused by managers and turnover intentions and has a partial mediating effect on the stress caused by work overload and career and pay. Exhaustion has a total mediation effect on the relationship between work overload and turnover intentions and a partial mediation effect when the predictor variable is stress caused by managers. The total mediation effect of exhaustion on the relationship between work overload and turnover intentions was expected, since, as the literature tells us, work overload is one of the factors that leads to more severe burnout (especially exhaustion) (Singh et al. 2020). These results align with the literature, which states that burnout mediates the relationship between occupational stress and turnover intentions (Han et al. 2016; Padmasundari 2019; Salama et al. 2022).

Hypothesis 5 was partially confirmed since only intrinsic motivation and integrated motivation have a negative and significant effect on turnover intentions. These results are in line with the literature since these two dimensions are part of what is known as autonomous motivation (Lopes et al. 2022; Kim 2018; Zheng et al. 2021). Autonomous motivation leads employees to engage in adaptive behaviors including turnover intentions (Fernet et al. 2017; Fernet et al. 2015; Trépanier et al. 2015).

Finally, there was evidence of the moderating effect of intrinsic motivation on the relationship between the stress caused by managers, colleagues, career and remuneration, and family problems. For participants with low intrinsic motivation, when compared

to participants with high intrinsic motivation, these dimensions of stress are relevant to boosting turnover intentions. These results align with the literature (Özbağ et al. 2014). High levels of intrinsic motivation allow for the effect of occupational stress (stress caused by managers, colleagues, career and pay, and family problems) on turnover to decrease in intensity.

The descriptive statistics of the variables under study showed that all the dimensions of occupational stress had a mean significantly higher than the central point of the scale, except for in stress caused by colleagues. The dimensions with the highest mean scores were the stress about career and pay, work overload, and working conditions. Those who showed the highest stress levels with career and pay were the employees of the customs offices, and the same was true of stress caused by working conditions. As for the stress caused by overwork, the highest levels were found among employees in the tax and customs offices.

Turnover intentions were also very high, indicating that many employees are considering leaving the Tax and Customs Authority soon. This may also be because the average age of these employees is very high. The employees with the highest levels of intention to leave are those working in the tax service, but it is precisely these employees who have the highest average age.

Burnout levels are also very high, with exhaustion levels higher than disengagement levels. It should be noted that among the dimensions of occupational stress, the one that has the strongest effect on burnout is work overload, which in turn, is one of the dimensions with the highest average. It was the employees in the customs offices who showed the highest levels of exhaustion. As for disengagement, it was the employees in the finance departments who showed the highest levels.

Concerning motivation, employees showed low levels of intrinsic motivation and introjected motivation but high levels of identified motivation. Employees in the finance department showed the lowest levels of intrinsic motivation, and those in the customs offices showed the highest levels of introjected motivation. Regarding identified motivation, those with the highest values are from the tax department. The low intrinsic and identified motivation levels in this study are in line with the results of the study conducted by Nishimura et al. (2021) on employees from the Portuguese Public Administration.

*Limitations*

Firstly, we have the limitation of the sampling process, which was nonprobabilistic, intentional and of the snowball type.

Secondly, the type of questionnaire used in this study, a self-report questionnaire, may have biased some of the participants' responses. However, several methodological and statistical recommendations were followed to reduce the impact of common method variance (Podsakoff et al. 2003).

Thirdly, it was a cross-sectional study, which does not allow us to establish causal relationships. It would be necessary to carry out a longitudinal study to test causal relationships.

## 6. Conclusions

This study is one of the first whose population is exclusively made up of employees of the Portuguese Tax and Customs Authority and which had an exceptional contribution from APIT to reach as many employees as possible to cover not only the mainland, but also the Autonomous Regions of Madeira and the Azores. It was confirmed that among the different departments of the Portuguese Tax and Customs Authority, the most critical departments are the tax offices and customs offices. The participants revealed high levels of stress, caused mainly by the problems with career progression and low pay. The fact that some careers have stagnated due to the crisis in Portugal since 2008 has caused many employees to feel a sense of injustice, and their stress levels have therefore increased. In recent years, we have witnessed several strikes by the tax and customs services in Portugal, where the main demands are based on the delay in regulating the revision of the careers

of tax workers, the growing degradation in the functioning and loss of authority of the AT, the performance evaluation system, and the "robotic functions that hinder the fight against fraud and tax evasion" (CNN Portugal 2021; Diário Digital de Castelo Branco 2023). Other factors that cause high levels of stress are work overload and working conditions. These high levels of stress lead to increased levels of burnout (Carod-Artal and Vázquez-Cabrera 2013; Galanakis et al. 2020; Khalid et al. 2020) and workers wanting to leave the organization, as some authors have pointed out (Jiang et al. 2022; Tziner et al. 2015). Those responsible for the Direct State Administration through the Ministry of Finance should be concerned with restoring career progression, reducing work overload, and improving the working conditions of these employees so that the levels of occupational stress are reduced, and subsequently, burnout and turnover intentions. Another concern is the high average age of employees, which is far too high (52.46 years) When you can join the civil service from 18, and the retirement age is 64 years and four months, this average age is high. In private organizations, the average is lower, often between 35 and 40 years old. We know that the Portuguese population is very elderly and that many young talents leave Portugal for other countries where job prospects are more attractive. However, the increase in the retirement age (66 years and four months) has also led to an increase in the average age of Portuguese public administration employees.

**Author Contributions:** Conceptualization, M.F., A.M. and F.R.; methodology, M.F. and A.M.; software, A.M.; validation, M.F., A.M. and F.R.; formal analysis, A.M.; investigation, A.M.; resources, A.M.; data curation, A.M.; writing—original draft preparation, M.F. and A.M.; writing—review and editing, M.F., A.M. and F.R.; visualization, A.M.; supervision, A.M.; project administration, M.F.; funding acquisition, A.M. All authors have read and agreed to the published version of the manuscript.

**Funding:** This research received no external funding.

**Institutional Review Board Statement:** Ethical review and approval were waived for this study since all participants (before answering the questionnaire) needed to read the informed consent portion and agree to it. This was the only way they could complete the questionnaire. Participants were informed about the purpose of the study, as well as that the results were confidential, as individual results would never be known and would only be analyzed in the set of all participants.

**Informed Consent Statement:** Informed consent was obtained from all subjects involved in the study.

**Data Availability Statement:** The data presented in this study are available upon request from the corresponding author. The data are not publicly available because in their informed consent, participants were informed that the data were confidential and that individual responses would never be known, as data analysis would be of all participants combined.

**Conflicts of Interest:** The authors declare no conflict of interest.

## Appendix A

**Table A1.** Moderating effect of identified motivation on the relationship between occupational stress and turnover intentions.

| Variables | B | SE | t | *p* | 95% CI |
|---|---|---|---|---|---|
| Users → turnover intentions ($R^2$ = 0.24; *p* < 0.001) | | | | | |
| Constant | 3.53 *** | 0.04 | 86.13 *** | <0.001 | [3.45, 3.61] |
| Users | 0.29 *** | 0.04 | 6.64 *** | <0.001 | [0.20, 0.37] |
| Identified motivation | −0.26 *** | 0.03 | −9.51 *** | <0.001 | [−0.32, −0.21] |
| U*IdM | 0.04 | 0.03 | 1.65 | 0.099 | [−0.08, 0.10] |
| Managers → turnover intentions ($R^2$ = 0.24; *p* < 0.001) | | | | | |
| Constant | 3.51 *** | 0.04 | 82.79 *** | <0.001 | [3.43, 3.60] |
| Managers | 0.28 *** | 0.04 | 4.04 *** | <0.001 | [0.20, 0.35] |
| Identified motivation | −0.23 *** | 0.03 | −8.01 *** | <0.001 | [−0.29, −0.17] |
| M*IdM | 0.01 | 0.02 | 0.24 | 0.810 | [−0.04, 0.05] |

**Table A1.** *Cont.*

| Variables | B | SE | t | p | 95% CI |
|---|---|---|---|---|---|
| Colleagues → turnover intentions ($R^2 = 0.23$; $p < 0.001$) | | | | | |
| Constant | 3.52 *** | 0.04 | 85.47 *** | <0.001 | [3.45, 3.61] |
| Colleagues | 0.23 *** | 0.03 | 5.96 *** | <0.001 | [0.16, 0.31] |
| Identified motivation | −0.27 *** | 0.03 | −9.71 *** | <0.001 | [−0.32, −0.21] |
| C*IdM | 0.04 | 0.02 | 1.65 | 0.100 | [−0.01, 0.09] |
| Overwork → turnover intentions ($R^2 = 0.26$; $p < 0.001$) | | | | | |
| Constant | 3.52 *** | 0.04 | 85.31 *** | <0.001 | [3.44, 3.60] |
| Overwork | 0.37 *** | 0.05 | 7.84 *** | <0.001 | [0.28, 0.47] |
| Identified motivation | −0.23 *** | 0.03 | −8.39 *** | <0.001 | [−0.29, −0.18] |
| O*IdM | 0.03 | 0.03 | 0.99 | 0.321 | [−0.03, 0.09] |
| Career and pay → turnover intentions ($R^2 = 0.21$; $p < 0.001$) | | | | | |
| Constant | 3.52 *** | 0.05 | 77.23 *** | <0.001 | [3.42, 3.60] |
| Career and pay | 0.22 *** | 0.05 | 4.07 *** | <0.001 | [0.11, 0.32] |
| Identified motivation | −0.25 *** | 0.03 | −7.90 *** | <0.001 | [−0.31, −0.19] |
| CP*IdM | 0.01 | 0.03 | 0.17 | 0.862 | [0.06, 0.07] |
| Family problems → Turnover Intentions ($R^2 = 0.22$; $p < 0.001$) | | | | | |
| Constant | 3.52 *** | 0.04 | 83.49 *** | <0.001 | [3.45, 3.61] |
| Family problems | 0.19 *** | 0.03 | 4.85 *** | <0.001 | [0.12, 0.27] |
| Identified motivation | −0.27 *** | 0.03 | −9.43 *** | <0.001 | [−0.32, −0.21] |
| FP*IdM | 0.03 | 0.02 | 1.37 | 0.173 | [−0.01, 0.08] |
| Work conditions→ turnover intentions ($R^2 = 0.21$; $p < 0.001$) | | | | | |
| Constant | 3.52 *** | 0.04 | 80.37 *** | <0.001 | [3.44, 3.61] |
| Work conditions | 0.21 *** | 0.05 | 4.53 *** | <0.001 | [0.12, 0.30] |
| Identified motivation | −0.26 *** | 0.03 | −8.60 *** | <0.001 | [−0.31, −0.20] |
| WC*IdM | 0.02 | 0.03 | 0.82 | 0.412 | [−0.03, 0.08] |

Note. *** $p < 0.001$.

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
