# Peer review of "Occupational Stress and Turnover Intentions in Employees of the Portuguese Tax and Customs Authority: Mediating Effect of Burnout and Moderating Effect of Motivation"

_admsci, doi:10.3390/admsci13120251_

Round 1

Reviewer 1 Report

Comments and Suggestions for Authors

The authors have followed the required  rigour of research. Like any output at  this  level.  What is the novelty? What new information re you bring to the table? What is the contribution of this work to literature apart from the normal rigor of work that give predestined results?

Kindly address this all around the body of work.

Comments on the Quality of English Language

The authors have followed the required  rigour of research. Like any output at  this  level.  What is the novelty? What new information re you bring to the table? What is the contribution of this work to literature apart from the normal rigor of work that give predestined results?

Kindly address this all around the body of work.

Author Response

Article

Occupational Stress and Turnover Intentions in employees of the Portuguese Tax and Customs Authority: Mediating effect of Burnout and moderating effect of Motivation

- Revision 1 -

Dear Reviewer,

Firstly, we would like to thank you and for taking the time and effort necessary to provide insightful guidance, which has contributed to improving this new version of the paper. We carefully considered the comments provided by the Reviewers. Herein, we explain how we revised the manuscript based on those comments and recommendations.

We appreciate your preliminary comments that will complement our work.

Comment 1: The authors have followed the required rigour of research. Like any output at this level.  What is the novelty? What new information re you bring to the table? What is the contribution of this work to literature apart from the normal rigor of work that give predestined results?

Kindly address this all around the body of work.

The innovation of this work is that it has inspectors from the Tax and Customs Authority as participants. It also resulted from a partnership with the Trade Union Association of Tax and Customs Inspection Professionals (APIT).

Another innovation is that motivation is a moderating variable in the relationship between occupational stress and turnover intentions.

In closing, we would like to thank you again for your comments. We hope that we have dealt with your suggestions satisfactorily and made all the adjustments requested, both in form and substance.

Yours sincerely,

On behalf of my co-authors,

References added to the manuscript:

(Ahn and Chaoyu, 2019) Ahn, Ji Young and Wang Chaoyu. 2019. Job stress and turnover intention revisited: evidence from KoreanFirms. Problems and Perspectives in Management, 17(4), 52-61 http://dx.doi.org/10.21511/ppm.17(4).2019.05

(Ajzen, 2012) Ajzen, Icek. 2012. The theory of planned behavior. In P. Lange, A. Kruglanski, & T. Higgins (Eds.), The handbook of theories of social psychology (pp. 438-459). London: Sage Publications.

(Breukelen et al., 2004) van Breukelen Wim, René van der Vlist, and Herman Steensma. 2004. Voluntary employee turnover: Combining variables from the ‘traditional’ turnover literature with the theory of planned behavior. Journal of Organizational Behavior, 25(7), 893–914. https://doi.org/10.1002/job.281

(Bester et al, 2015) Bester, Janie, Marius Wilhelm Stander, and Llewellyn Ellardus Van Zyl. 2015. Leadership empowering behaviour, psychological empowerment, organisational citizenship behaviours and turnover intention in a manufacturing division: Original research. SA Journal of Industrial Psychology, 41(1), 1–14.

(CNN Portugal, 2021) CNN Portugal. (2021) [Accessed November 22, 2023]  https://cnnportugal.iol.pt/trabalhadores/autoridade-tributaria-e-aduaneira/greve-dos-trabalhores-dos-impostos-e-alfandegas-encerra-70-dos-servicos-diz-sindicato/20271231/61a8eb890cf2cc58e7d5decc

(Deci et al., 2017) Deci, Eduard L., Anja H. Olafsen, and Richard Ryan. 2017). Self-determination theory in work organizations: The state of a science. Annual Review of Organizational Psychology and Organizational Behavior, 4(1), 19–43.

(Deci and Ryan, 2008) Deci, Eduard L., and Richard Ryan. (2008). Self-determination theory: A macrotheory of human motivation, development, and health. Canadian Psychology, 49(3),182–185.

(Diário Digital de Castelo Branco, 2023) Diário Digital de Castelo Branco (2023) [Accessed November 22, 2023] https://www.diariodigitalcastelobranco.pt/noticia/62547/distrito-de-castelo-branco-protesta-contra-degradacao-da-autoridade-tributaria-e-aduaneira--

(Fernet et al., 2017) Fernet, Claude, Sarah-Geneviève Trépanier, Mireille Demers, and Stéphanie Austin. 2017. Motivational pathways of occupational and organizational turnover intention among newly registered nurses in Canada. Nursing outlook, 65(4), 444–454. https://doi.org/10.1016/j.outlook.2017.05.008

(Fernet et al., 2015) Fernet, Claude, Sarah-Geneviève Trépanier, Stéphanie Austin, Marylène Gagné, and Jacques Forest. 2015. Transformational leadership and optimal functioning at work: On the mediating role of employees' perceived job characteristics and motivation. Work & Stress, 29(1), 11–31. https://doi.org/10.1080/02678373.2014.1003998

(Galanakis et al., 2020) Galanakis, Michael, Evangelia Alexandri, Kalliopi Kika, Xristofili Lelekanou, Margarita Papantonopoulou, Dimitra Stougiannou, Melina Tzani. (2020).  What Is the Source of Occupational Stress and Burnout? Psychology, 11(05), 647−664. https://doi.org/10.4236/psych.2020.115044

(Gautam and Gautam, 2022) Gautam, Dhruba and Prakash Kumar Gautam. 2022. Occupational stress for employee turnover intention: mediation effect of service climate and emotion regulation. Asia-Pacific Journal of Business Administration, Vol. ahead-of-print No. ahead-of-print. https://doi.org/10.1108/APJBA-02-2021-0056

(Gok et al., 2017) Gok, Ozge Adan, Yilmaz Akgunduz, and Ceylan Alkan. 2017. The effects of job stress and perceived organizationalsupport on turnover intentions of hotel employees. Journal of Tourismology, 3(2), 23-32.

(Huang et al., 2003) Huang In-Chung, Chih-Hsun Jason Chuang, Hao-Chieh. 2003. The role of burnout in the relationship between perceptions off organizational politics and turnover intentions. Public Personnel Management, 32(4), 519–531. https://doi.org/10.1177/009102600303200404

(Imeokparia and Ediagbonya , 2013) Imeokparia, Patience Osebhakhomen, and Kennedy Ediagbonya. 2013. Stress management: An approach to ensuring high academic performance of business education students. European Journal of Educational Studies 5, 167–76

(Islam et al., 2019). Islam, Nazrul, Ekhtear Ahmed Zeesan, Debanik Chakraborty, Md. Nafizur Rahman, Syed Istiak Uddin Ahmed, Nowshin Nower, and Toufiq Nazrul. 2019. Relationship between Job Stress and the Turnover Intention of Private Sector Bank Employees in Bangladesh. International Business Research. DOI:10.5539/ibr.v12n8p133

(Jabutay and Rungruang, 2020) Jabutay, Felicito, and Parisa Rungruang. 2020. Turnover intent of new workers: social exchangeperspectives. Asia-Pacific Journal of Business Administration, 13 (1), 60-79.

(Khalid et al., 2020) Khalid, Arslan, Fang Pan, Ping Li, Wei Wang, and Abdul Sattar Ghaffari.  2020.  The impact of occupational stress on job burnout among bank employees in Pakistan, with psychological capital as a mediator. Frontiers in Public Health,7, 410−419.https://doi.org/10.3389/fpubh.2019.00410

Kusy, Kevin, Sara O’Leary-Driscoll. 2020. Combat negativity, exhaustion, and burnout. [Accessed November 22, 2023]  

https://digitalcommons.imsa.edu/cgi/viewcontent.cgi?article=1398&=&context=proflearningday&=&sei

(Lee, 2019) Lee, Yee Hoon. 2019. Emotional labor, teacher burnout, and turnover intention in high-school physical education teaching. European Physical Education Review, 25(1), 236–253. DOI: 10.1177/1356336X17719559

(Lee and Song, 2020) Lee, Jung-Hoon, and Yeoungsuk Song. 2020. Mixed method research investigating turnover intention with ICUnurses. Journal of Korean Academy of Fundamentals of Nursing, 27(2), 153-163

(Ordem dos Psicólogos, 2023) Ordem dos Psicólogos. 2023. II Relatório do custo do stresse e dos problemas de saúde psicológica no trabalho, em Portugal [Accessed November 22, 2023]. https://www.ordemdospsicologos.pt/pt/noticia/4466

(Padmasundari, 2019) Padmasundari S. (2019). A psychological study on burnout among school teachers. Indian Journal of Applied Research, 9(12), 22–23.

(Regts and Molleman, 2013) Regts Gerdian, Eric Molleman. 2013. To leave or not to leave: When receiving interpersonal citizenship behavior influences an employee’s turnover intention. Human Relations, 66(2), 193–218. https://doi.org/10.1177/0018726712454311

(Singh et al., 2020) Singh, Ankit, Ajeya Jha, and Shankar Purbey. 2020. Identification of Measures Affecting Job Satisfaction and Levels of Perceived Stress and Burnout among Home Health Nurses of a Developing Asian Country. Hospital Topics, 1−11.https://doi.org/10.1080/00185868.2020.1830009

(Trépanier et al., 2015) Trépanier, Sarah-Geneviève, Jacques Forest, Claude Fernet, and Stéphanie Austin. 2015. On the psychological and motivational processes linking job characteristics to employee functioning: Insights from self-determination theory. Work & Stress, 29(3), 286–305. https://doi.org/10.1080/02678373.2015.1074957

(Urbanaviciute et al., 2018) Urbanaviciute Ieva, Jurjita Lazauskaite-Zabielske, Tine Vander Elst, Hans De Witte. 2018. Qualitative job insecurity and turnover intention: The mediating role of basic psychological needs in public and private sectors. Career Development International, 23(3), 274–290. https://doi.org/10.1108/cdi-07-2017-0117

Wang, Yau-De, Chyan Yang, Wang Kuei-Ying. 2012. Comparing public and private employees’ job satisfaction and turnover. Public Personnel Management, 41(3), 557–573. https://doi.org/10.1177/009102601204100310

(Yao et al, 2019) Yao, Bo-chen, Ling-bing Meng, Meng-lei Hao, Yuan-meng Zhang, Tao Gong, and Zhi-gang Guo. 2019. Chronic stress: A critical risk factor for atherosclerosis. Journal of International Medical Research 47: 1429–40.

Reviewer 2 Report

Comments and Suggestions for Authors

Dear authors,

one of the papers big weaknesses is a luck of a comprehensive literature review, identifying the research gaps and statement of contribution of the conducted research. 

The literature review is poor and presented with  focus on studies /authors, not on synthesis and integration.  

Occupational stress is a crucial concept in your article. You change dimensions in some hypothesis without further explanation. 

Consider developing  the section 2. Theoretical Framework and Research Hypotheses. Now this section is limited while in the section 3.  Materials and Methods different descriptions of theoretical framework are  to find.

I have a wondering about Conclusions:

You state: “Another concern is the high average age 560 of employees, which is far too high” . Way it is a problem? And which age and on which bases you classify as “far too high”.

Some issues about in the text -citations: 

page number of the quotation is missing

 you mention several studies and cite only few  (for example line 110). 
“Several studies report that burnout has a mediating effect on the relationship 110 between occupational stress and turnover intentions (Han et al., 2016; Salama et al., 2022”).

Due to my  limited competence I am not giving you feedback on the quality of the analysis. 

Comments on the Quality of English Language

  English level  in your paper is very poor

some examples: 

·       see line 34,35

·       To better understand the concept of motivation, various researchers have proposed  theories on motivation to understand its origin and influence on human behavior (Silva  and Gomes, 2009)” (line 125,126)

·       “Thus, the authors present a multidi-128 mensional view of motivation based on three points: amotivation, which is contrary to  motivation, i.e. it is the absence of motivation for a particular task; intrinsic motivation,  characterized as a pleasant and interesting feeling, linked to intrinsic motivation, we have identified motivation, referring to performing a task because one identifies with it and it  has a value and/or meaning (Grohmann et al., 2013); extrinsic motivation, characterized  as a feeling that comes from the environment in order to avoid criticism and promote the  self-esteem of others (Gagné et al., 2015)”. (line 128-134)  I

You should improve narrative in-text citations

·       for example “according to” is overused, see for example section 2.2; moreover formulation  such as “to” (line 30),   “for” (line 101), “from the perspective of” (line 29, 487).

Author Response

Article

Occupational Stress and Turnover Intentions in employees of the Portuguese Tax and Customs Authority: Mediating effect of Burnout and moderating effect of Motivation

- Revision 1 -

Dear Reviewer,

Firstly, we would like to thank you and for taking the time and effort necessary to provide insightful guidance, which has contributed to improving this new version of the paper. We carefully considered the comments provided by the Reviewers. Herein, we explain how we revised the manuscript based on those comments and recommendations.

We appreciate your comments that will complement our work.

We are very thankfully for all reviewer 2 interesting insights.

Comment 1: The literature review is poor and presented with focus on studies /authors, not on synthesis and integration.

The literature review has been improved. We hope it is in line with what you suggested.

Comment 2: Occupational stress is a crucial concept in your article. You change dimensions in some hypothesis without further explanation.

You are right, and the correction has been made.

Comment 3: Consider developing the section 2. Theoretical Framework and Research Hypotheses. Now this section is limited while in the section 3.  Materials and Methods different descriptions of theoretical framework are to find.

Section 2 has also been improved, as you suggested.

Comment 4: You state: “Another concern is the high average age 560 of employees, which is far too high”. Way it is a problem? And which age and on which bases you classify as “far too high”.

If we compare public administration with private organizations, the average age is very high. As such, these employees are tired, their salaries are relatively low, their stress and burnout levels are high, and they want to leave the organization.

Comment 5: page number of the quotation is missing

There are no direct citations in the text. I have just added one when I refer to a news item from CNN Portugal. And I should only add the page when the quote is direct.

Comment 6: You mention several studies and cite only few  (for example line 110).

“Several studies report that burnout has a mediating effect on the relationship 110 between occupational stress and turnover intentions (Han et al., 2016; Salama et al., 2022”).

I tried to improve this aspect and add some studies.

Comment 7: English level in your paper is very poor.

some examples:

 see line 34,35

“To better understand the concept of motivation, various researchers have proposed  theories on motivation to understand its origin and influence on human behavior (Silva  and Gomes, 2009)” (line 125,126)

“Thus, the authors present a multidi-128 mensional view of motivation based on three points: amotivation, which is contrary to  motivation, i.e. it is the absence of motivation for a particular task; intrinsic motivation,  characterized as a pleasant and interesting feeling, linked to intrinsic motivation, we have identified motivation, referring to performing a task because one identifies with it and it  has a value and/or meaning (Grohmann et al., 2013); extrinsic motivation, characterized  as a feeling that comes from the environment in order to avoid criticism and promote the  self-esteem of others (Gagné et al., 2015)”. (line 128-134) 

I also tried to improve my English.

Comment 8: You should improve narrative in-text citations:

  • for example “according to” is overused, see for example section 2.2; moreover formulation such as “to” (line 30),   “for” (line 101), “from the perspective of” (line 29, 487).

I also tried to improve the narrative quotes in the text.

In closing, we would like to thank you again for your comments. We hope that we have dealt with your suggestions satisfactorily and made all the adjustments requested, both in form and substance.

Yours sincerely,

On behalf of my co-authors,

References added to the manuscript:

(Ahn and Chaoyu, 2019) Ahn, Ji Young and Wang Chaoyu. 2019. Job stress and turnover intention revisited: evidence from KoreanFirms. Problems and Perspectives in Management, 17(4), 52-61 http://dx.doi.org/10.21511/ppm.17(4).2019.05

(Ajzen, 2012) Ajzen, Icek. 2012. The theory of planned behavior. In P. Lange, A. Kruglanski, & T. Higgins (Eds.), The handbook of theories of social psychology (pp. 438-459). London: Sage Publications.

(Breukelen et al., 2004) van Breukelen Wim, René van der Vlist, and Herman Steensma. 2004. Voluntary employee turnover: Combining variables from the ‘traditional’ turnover literature with the theory of planned behavior. Journal of Organizational Behavior, 25(7), 893–914. https://doi.org/10.1002/job.281

(Bester et al, 2015) Bester, Janie, Marius Wilhelm Stander, and Llewellyn Ellardus Van Zyl. 2015. Leadership empowering behaviour, psychological empowerment, organisational citizenship behaviours and turnover intention in a manufacturing division: Original research. SA Journal of Industrial Psychology, 41(1), 1–14.

(CNN Portugal, 2021) CNN Portugal. (2021) [Accessed November 22, 2023]  https://cnnportugal.iol.pt/trabalhadores/autoridade-tributaria-e-aduaneira/greve-dos-trabalhores-dos-impostos-e-alfandegas-encerra-70-dos-servicos-diz-sindicato/20271231/61a8eb890cf2cc58e7d5decc

(Deci et al., 2017) Deci, Eduard L., Anja H. Olafsen, and Richard Ryan. 2017). Self-determination theory in work organizations: The state of a science. Annual Review of Organizational Psychology and Organizational Behavior, 4(1), 19–43.

(Deci and Ryan, 2008) Deci, Eduard L., and Richard Ryan. (2008). Self-determination theory: A macrotheory of human motivation, development, and health. Canadian Psychology, 49(3),182–185.

(Diário Digital de Castelo Branco, 2023) Diário Digital de Castelo Branco (2023) [Accessed November 22, 2023] https://www.diariodigitalcastelobranco.pt/noticia/62547/distrito-de-castelo-branco-protesta-contra-degradacao-da-autoridade-tributaria-e-aduaneira--

(Fernet et al., 2017) Fernet, Claude, Sarah-Geneviève Trépanier, Mireille Demers, and Stéphanie Austin. 2017. Motivational pathways of occupational and organizational turnover intention among newly registered nurses in Canada. Nursing outlook, 65(4), 444–454. https://doi.org/10.1016/j.outlook.2017.05.008

(Fernet et al., 2015) Fernet, Claude, Sarah-Geneviève Trépanier, Stéphanie Austin, Marylène Gagné, and Jacques Forest. 2015. Transformational leadership and optimal functioning at work: On the mediating role of employees' perceived job characteristics and motivation. Work & Stress, 29(1), 11–31. https://doi.org/10.1080/02678373.2014.1003998

(Galanakis et al., 2020) Galanakis, Michael, Evangelia Alexandri, Kalliopi Kika, Xristofili Lelekanou, Margarita Papantonopoulou, Dimitra Stougiannou, Melina Tzani. (2020).  What Is the Source of Occupational Stress and Burnout? Psychology, 11(05), 647−664. https://doi.org/10.4236/psych.2020.115044

(Gautam and Gautam, 2022) Gautam, Dhruba and Prakash Kumar Gautam. 2022. Occupational stress for employee turnover intention: mediation effect of service climate and emotion regulation. Asia-Pacific Journal of Business Administration, Vol. ahead-of-print No. ahead-of-print. https://doi.org/10.1108/APJBA-02-2021-0056

(Gok et al., 2017) Gok, Ozge Adan, Yilmaz Akgunduz, and Ceylan Alkan. 2017. The effects of job stress and perceived organizational support on turnover intentions of hotel employees. Journal of Tourismology, 3(2), 23-32.

(Huang et al., 2003) Huang In-Chung, Chih-Hsun Jason Chuang, Hao-Chieh. 2003. The role of burnout in the relationship between perceptions off organizational politics and turnover intentions. Public Personnel Management, 32(4), 519–531. https://doi.org/10.1177/009102600303200404

(Imeokparia and Ediagbonya , 2013) Imeokparia, Patience Osebhakhomen, and Kennedy Ediagbonya. 2013. Stress management: An approach to ensuring high academic performance of business education students. European Journal of Educational Studies 5, 167–76

(Islam et al., 2019). Islam, Nazrul, Ekhtear Ahmed Zeesan, Debanik Chakraborty, Md. Nafizur Rahman, Syed Istiak Uddin Ahmed, Nowshin Nower, and Toufiq Nazrul. 2019. Relationship between Job Stress and the Turnover Intention of Private Sector Bank Employees in Bangladesh. International Business Research. DOI:10.5539/ibr.v12n8p133

(Jabutay and Rungruang, 2020) Jabutay, Felicito, and Parisa Rungruang. 2020. Turnover intent of new workers: social exchangeperspectives. Asia-Pacific Journal of Business Administration, 13 (1), 60-79.

(Khalid et al., 2020) Khalid, Arslan, Fang Pan, Ping Li, Wei Wang, and Abdul Sattar Ghaffari.  2020.  The impact of occupational stress on job burnout among bank employees in Pakistan, with psychological capital as a mediator. Frontiers in Public Health,7, 410−419.https://doi.org/10.3389/fpubh.2019.00410

Kusy, Kevin, Sara O’Leary-Driscoll. 2020. Combat negativity, exhaustion, and burnout. [Accessed November 22, 2023]  

https://digitalcommons.imsa.edu/cgi/viewcontent.cgi?article=1398&=&context=proflearningday&=&sei

(Lee, 2019) Lee, Yee Hoon. 2019. Emotional labor, teacher burnout, and turnover intention in high-school physical education teaching. European Physical Education Review, 25(1), 236–253. DOI: 10.1177/1356336X17719559

(Lee and Song, 2020) Lee, Jung-Hoon, and Yeoungsuk Song. 2020. Mixed method research investigating turnover intention with ICUnurses. Journal of Korean Academy of Fundamentals of Nursing, 27(2), 153-163

(Ordem dos Psicólogos, 2023) Ordem dos Psicólogos. 2023. II Relatório do custo do stresse e dos problemas de saúde psicológica no trabalho, em Portugal [Accessed November 22, 2023]. https://www.ordemdospsicologos.pt/pt/noticia/4466

(Padmasundari, 2019) Padmasundari S. (2019). A psychological study on burnout among school teachers. Indian Journal of Applied Research, 9(12), 22–23.

(Regts and Molleman, 2013) Regts Gerdian, Eric Molleman. 2013. To leave or not to leave: When receiving interpersonal citizenship behavior influences an employee’s turnover intention. Human Relations, 66(2), 193–218. https://doi.org/10.1177/0018726712454311

(Singh et al., 2020) Singh, Ankit, Ajeya Jha, and Shankar Purbey. 2020. Identification of Measures Affecting Job Satisfaction and Levels of Perceived Stress and Burnout among Home Health Nurses of a Developing Asian Country. Hospital Topics, 1−11.https://doi.org/10.1080/00185868.2020.1830009

(Trépanier et al., 2015) Trépanier, Sarah-Geneviève, Jacques Forest, Claude Fernet, and Stéphanie Austin. 2015. On the psychological and motivational processes linking job characteristics to employee functioning: Insights from self-determination theory. Work & Stress, 29(3), 286–305. https://doi.org/10.1080/02678373.2015.1074957

(Urbanaviciute et al., 2018) Urbanaviciute Ieva, Jurjita Lazauskaite-Zabielske, Tine Vander Elst, Hans De Witte. 2018. Qualitative job insecurity and turnover intention: The mediating role of basic psychological needs in public and private sectors. Career Development International, 23(3), 274–290. https://doi.org/10.1108/cdi-07-2017-0117

Wang, Yau-De, Chyan Yang, Wang Kuei-Ying. 2012. Comparing public and private employees’ job satisfaction and turnover. Public Personnel Management, 41(3), 557–573. https://doi.org/10.1177/009102601204100310

(Yao et al, 2019) Yao, Bo-chen, Ling-bing Meng, Meng-lei Hao, Yuan-meng Zhang, Tao Gong, and Zhi-gang Guo. 2019. Chronic stress: A critical risk factor for atherosclerosis. Journal of International Medical Research 47: 1429–40.

Reviewer 3 Report

Comments and Suggestions for Authors

  • The manuscript studies (a) the impact of occupational stress on turnover intentions and (b) the effects of burnout levels and motivation the above relationship.  In general, the paper is a very interesting and novel approach to the critical issue of occupational stress and burnout. More specifically, the pertinent topic is innovative enough. In fact there is not much literature on the Portuguese case, besides the latter has not been studied to a high extend in the past. The manuscript contributes to that field as it addresses a special public service in Portugal (Tax and Customs Authority) with several features of the workforce (age, working conditions, brain drain).     
  • This paper is scientifically sound, and the ethics statement is adequate. The paper fits the scope of the journal. The significance of the paper is high enough. The quality of the paper is good enough, as it has been written appropriately. The manuscript will be interesting for the readers.
  • Similar studies that have been contacted in Southern European countries with similar climate i.e., Greece, Italy, Spain etc. could provide a useful comparative terrain. Authors should describe clearly their sample selection methods and explain how they assured that nobody completed the same questionnaire twice. More specifically, authors should clarify the way that they collected their sample in order to enhance validity and reliability of their research project and assure that the sample reflects the pertinent population. 
  • The overall merit is positive, the publication will bring about benefits in knowledge. The conclusions are interesting and well-connected with the relevant evidence. Last, the references have been relevant and valid. The references are relevant, but they should be updated with sources after 2019.

Author Response

Article

Occupational Stress and Turnover Intentions in employees of the Portuguese Tax and Customs Authority: Mediating effect of Burnout and moderating effect of Motivation

- Revision 1 -

Dear Reviewer,

Firstly, we would like to thank you and for taking the time and effort necessary to provide insightful guidance, which has contributed to improving this new version of the paper. We carefully considered the comments provided by the Reviewers. Herein, we explain how we revised the manuscript based on those comments and recommendations.

We appreciate your preliminary comments that will complement our work.

Comment 1: The manuscript studies (a) the impact of occupational stress on turnover intentions and (b) the effects of burnout levels and motivation the above relationship.  In general, the paper is a very interesting and novel approach to the critical issue of occupational stress and burnout. More specifically, the pertinent topic is innovative enough. In fact there is not much literature on the Portuguese case, besides the latter has not been studied to a high extend in the past. The manuscript contributes to that field as it addresses a special public service in Portugal (Tax and Customs Authority) with several features of the workforce (age, working conditions, brain drain).    

This paper is scientifically sound, and the ethics statement is adequate. The paper fits the scope of the journal. The significance of the paper is high enough. The quality of the paper is good enough, as it has been written appropriately. The manuscript will be interesting for the readers.

Similar studies that have been contacted in Southern European countries with similar climate i.e., Greece, Italy, Spain etc. could provide a useful comparative terrain. Authors should describe clearly their sample selection methods and explain how they assured that nobody completed the same questionnaire twice. More specifically, authors should clarify the way that they collected their sample in order to enhance validity and reliability of their research project and assure that the sample reflects the pertinent population.

The overall merit is positive, the publication will bring about benefits in knowledge. The conclusions are interesting and well-connected with the relevant evidence. Last, the references have been relevant and valid. The references are relevant, but they should be updated with sources after 2019.

The data collection process was reformulated as information was added. This work results from a partnership with the Trade Union Association of Tax and Customs Inspection Professionals (APIT). It was one of the APIT leaders who distributed the questionnaire link.

The references have been updated.

In closing, we would like to thank you again for your comments. We hope that we have dealt with your suggestions satisfactorily and made all the adjustments requested, both in form and substance.

Yours sincerely,

On behalf of my co-authors,

References added to the manuscript:

(Ahn and Chaoyu, 2019) Ahn, Ji Young and Wang Chaoyu. 2019. Job stress and turnover intention revisited: evidence from KoreanFirms. Problems and Perspectives in Management, 17(4), 52-61 http://dx.doi.org/10.21511/ppm.17(4).2019.05

(Ajzen, 2012) Ajzen, Icek. 2012. The theory of planned behavior. In P. Lange, A. Kruglanski, & T. Higgins (Eds.), The handbook of theories of social psychology (pp. 438-459). London: Sage Publications.

(Breukelen et al., 2004) van Breukelen Wim, René van der Vlist, and Herman Steensma. 2004. Voluntary employee turnover: Combining variables from the ‘traditional’ turnover literature with the theory of planned behavior. Journal of Organizational Behavior, 25(7), 893–914. https://doi.org/10.1002/job.281

(Bester et al, 2015) Bester, Janie, Marius Wilhelm Stander, and Llewellyn Ellardus Van Zyl. 2015. Leadership empowering behaviour, psychological empowerment, organisational citizenship behaviours and turnover intention in a manufacturing division: Original research. SA Journal of Industrial Psychology, 41(1), 1–14.

(CNN Portugal, 2021) CNN Portugal. (2021) [Accessed November 22, 2023]  https://cnnportugal.iol.pt/trabalhadores/autoridade-tributaria-e-aduaneira/greve-dos-trabalhores-dos-impostos-e-alfandegas-encerra-70-dos-servicos-diz-sindicato/20271231/61a8eb890cf2cc58e7d5decc

(Deci et al., 2017) Deci, Eduard L., Anja H. Olafsen, and Richard Ryan. 2017). Self-determination theory in work organizations: The state of a science. Annual Review of Organizational Psychology and Organizational Behavior, 4(1), 19–43.

(Deci and Ryan, 2008) Deci, Eduard L., and Richard Ryan. (2008). Self-determination theory: A macrotheory of human motivation, development, and health. Canadian Psychology, 49(3),182–185.

(Diário Digital de Castelo Branco, 2023) Diário Digital de Castelo Branco (2023) [Accessed November 22, 2023] https://www.diariodigitalcastelobranco.pt/noticia/62547/distrito-de-castelo-branco-protesta-contra-degradacao-da-autoridade-tributaria-e-aduaneira--

(Fernet et al., 2017) Fernet, Claude, Sarah-Geneviève Trépanier, Mireille Demers, and Stéphanie Austin. 2017. Motivational pathways of occupational and organizational turnover intention among newly registered nurses in Canada. Nursing outlook, 65(4), 444–454. https://doi.org/10.1016/j.outlook.2017.05.008

(Fernet et al., 2015) Fernet, Claude, Sarah-Geneviève Trépanier, Stéphanie Austin, Marylène Gagné, and Jacques Forest. 2015. Transformational leadership and optimal functioning at work: On the mediating role of employees' perceived job characteristics and motivation. Work & Stress, 29(1), 11–31. https://doi.org/10.1080/02678373.2014.1003998

(Galanakis et al., 2020) Galanakis, Michael, Evangelia Alexandri, Kalliopi Kika, Xristofili Lelekanou, Margarita Papantonopoulou, Dimitra Stougiannou, Melina Tzani. (2020).  What Is the Source of Occupational Stress and Burnout? Psychology, 11(05), 647−664. https://doi.org/10.4236/psych.2020.115044

(Gautam and Gautam, 2022) Gautam, Dhruba and Prakash Kumar Gautam. 2022. Occupational stress for employee turnover intention: mediation effect of service climate and emotion regulation. Asia-Pacific Journal of Business Administration, Vol. ahead-of-print No. ahead-of-print. https://doi.org/10.1108/APJBA-02-2021-0056

(Gok et al., 2017) Gok, Ozge Adan, Yilmaz Akgunduz, and Ceylan Alkan. 2017. The effects of job stress and perceived organizationalsupport on turnover intentions of hotel employees. Journal of Tourismology, 3(2), 23-32.

(Huang et al., 2003) Huang In-Chung, Chih-Hsun Jason Chuang, Hao-Chieh. 2003. The role of burnout in the relationship between perceptions off organizational politics and turnover intentions. Public Personnel Management, 32(4), 519–531. https://doi.org/10.1177/009102600303200404

(Imeokparia and Ediagbonya , 2013) Imeokparia, Patience Osebhakhomen, and Kennedy Ediagbonya. 2013. Stress management: An approach to ensuring high academic performance of business education students. European Journal of Educational Studies 5, 167–76

(Islam et al., 2019). Islam, Nazrul, Ekhtear Ahmed Zeesan, Debanik Chakraborty, Md. Nafizur Rahman, Syed Istiak Uddin Ahmed, Nowshin Nower, and Toufiq Nazrul. 2019. Relationship between Job Stress and the Turnover Intention of Private Sector Bank Employees in Bangladesh. International Business Research. DOI:10.5539/ibr.v12n8p133

(Jabutay and Rungruang, 2020) Jabutay, Felicito, and Parisa Rungruang. 2020. Turnover intent of new workers: social exchangeperspectives. Asia-Pacific Journal of Business Administration, 13 (1), 60-79.

(Khalid et al., 2020) Khalid, Arslan, Fang Pan, Ping Li, Wei Wang, and Abdul Sattar Ghaffari.  2020.  The impact of occupational stress on job burnout among bank employees in Pakistan, with psychological capital as a mediator. Frontiers in Public Health,7, 410−419.https://doi.org/10.3389/fpubh.2019.00410

Kusy, Kevin, Sara O’Leary-Driscoll. 2020. Combat negativity, exhaustion, and burnout. [Accessed November 22, 2023]  

https://digitalcommons.imsa.edu/cgi/viewcontent.cgi?article=1398&=&context=proflearningday&=&sei

(Lee, 2019) Lee, Yee Hoon. 2019. Emotional labor, teacher burnout, and turnover intention in high-school physical education teaching. European Physical Education Review, 25(1), 236–253. DOI: 10.1177/1356336X17719559

(Lee and Song, 2020) Lee, Jung-Hoon, and Yeoungsuk Song. 2020. Mixed method research investigating turnover intention with ICUnurses. Journal of Korean Academy of Fundamentals of Nursing, 27(2), 153-163

(Ordem dos Psicólogos, 2023) Ordem dos Psicólogos. 2023. II Relatório do custo do stresse e dos problemas de saúde psicológica no trabalho, em Portugal [Accessed November 22, 2023]. https://www.ordemdospsicologos.pt/pt/noticia/4466

(Padmasundari, 2019) Padmasundari S. (2019). A psychological study on burnout among school teachers. Indian Journal of Applied Research, 9(12), 22–23.

(Regts and Molleman, 2013) Regts Gerdian, Eric Molleman. 2013. To leave or not to leave: When receiving interpersonal citizenship behavior influences an employee’s turnover intention. Human Relations, 66(2), 193–218. https://doi.org/10.1177/0018726712454311

(Singh et al., 2020) Singh, Ankit, Ajeya Jha, and Shankar Purbey. 2020. Identification of Measures Affecting Job Satisfaction and Levels of Perceived Stress and Burnout among Home Health Nurses of a Developing Asian Country. Hospital Topics, 1−11.https://doi.org/10.1080/00185868.2020.1830009

(Trépanier et al., 2015) Trépanier, Sarah-Geneviève, Jacques Forest, Claude Fernet, and Stéphanie Austin. 2015. On the psychological and motivational processes linking job characteristics to employee functioning: Insights from self-determination theory. Work & Stress, 29(3), 286–305. https://doi.org/10.1080/02678373.2015.1074957

(Urbanaviciute et al., 2018) Urbanaviciute Ieva, Jurjita Lazauskaite-Zabielske, Tine Vander Elst, Hans De Witte. 2018. Qualitative job insecurity and turnover intention: The mediating role of basic psychological needs in public and private sectors. Career Development International, 23(3), 274–290. https://doi.org/10.1108/cdi-07-2017-0117

Wang, Yau-De, Chyan Yang, Wang Kuei-Ying. 2012. Comparing public and private employees’ job satisfaction and turnover. Public Personnel Management, 41(3), 557–573. https://doi.org/10.1177/009102601204100310

(Yao et al., 2019) Yao, Bo-chen, Ling-bing Meng, Meng-lei Hao, Yuan-meng Zhang, Tao Gong, and Zhi-gang Guo. 2019. Chronic stress: A critical risk factor for atherosclerosis. Journal of International Medical Research 47: 1429–40.

Round 2

Reviewer 2 Report

Comments and Suggestions for Authors

Dear authors,

you  have revised the paper, mostly the literature review.

Unfortunately, there are still missing words and not well formulated in text- citation (please see examples below and attached file). I have also some comments about literature and a wondering about age. 

·      I am not assessing the quality of the whole literature review, but I have identified two studies, which should be mentioned. One study actually about tax office workers, 

Issever, H., Ozdilli, K., Altunkaynak, O., Onen, L., & Disci, R. (2008). Depression in tax office workers in Istanbul and its affecting factors. Indoor and Built Environment, 17(5), 414-420.

and another study about motivation in the public sector which should be involved in the following hypothesis formulation  2.5. Motivation at work and turnover intentions

Kim, J. (2018). The contrary effects of intrinsic and extrinsic motivations on burnout and turnover intention in the public sector. International journal of manpower, 39(3), 486-500.

·      Moreover, one more wondering from my side, which I have missed in the first round: 

If you have data on respondents' age, why isn't this included as an independent variable in your regressions? Currently the reader must wonder whether all of the relationships with turnover intentions are better explained by a separate variable--approaching retirement age (line 578:) 

·      indentations for paragraphs change from line 209 throughout the rest of the article

·      4: typo:  moderating

·      17 clarity: to try to minimize their causes

·      29: 20212???

·      567 clarity: "...mean that..." is unclear. Consider "...correlate with..." or "...allow..." 

·      536: repeated sentence

·      589 missing word: "...high levels of ??? motivation."

·      629 missing word: 64 years and 4 months

·      consider changing "stress with..." to either "stress from..." or "stress caused by..." or "stress stemming from..." (stress with managers can mean that managers feel stressed)

Author Response

Article

Occupational Stress and Turnover Intentions in employees of the Portuguese Tax and Customs Authority: Mediating effect of Burnout and moderating effect of Motivation

- Revision 2 -

Dear Reviewer,

Firstly, we would like to thank you and for taking the time and effort necessary to provide insightful guidance, which has contributed to improving this new version of the paper. We carefully considered the comments provided by the Reviewers. Herein, we explain how we revised the manuscript based on those comments and recommendations.

We appreciate your comments that will complement our work.

We are very thankful for all the interesting insights.

Comment 1: ·      I am not assessing the quality of the whole literature review, but I have identified two studies, which should be mentioned. One study actually about tax office workers,

Issever, H., Ozdilli, K., Altunkaynak, O., Onen, L., & Disci, R. (2008). Depression in tax office workers in Istanbul and its affecting factors. Indoor and Built Environment, 17(5), 414-420.

This study has been added to the subchapter "2.1 Job Occupational Stress and Turnover Intentions."

Comment 2: and another study about motivation in the public sector which should be involved in the following hypothesis formulation  2.5. Motivation at work and turnover intentions.

This study has been added to the subchapter “2.5. Motivation at work and turnover intentions

Comment 3: Moreover, one more wondering from my side, which I have missed in the first round:

If you have data on respondents' age, why isn't this included as an independent variable in your regressions? Currently the reader must wonder whether all of the relationships with turnover intentions are better explained by a separate variable--approaching retirement age (line 578:)

I added age to the correlation table, which did not correlate significantly with any dependent variables. I mentioned this situation at the beginning of the hypothesis description, so I didn't consider age as a control variable.

Comment 4: indentations for paragraphs change from line 209 throughout the rest of the article

You are right, and the correction has been made.

Comment 5: ·      4: typo:  moderating

It has been corrected.

Comment 6: 17 clarity: to try to minimize their causes

The sentence has been reworded to make it more explicit.

Comment 7: 29: 20212???

It has been corrected.

Comment 8: 567 clarity: "...mean that..." is unclear. Consider "...correlate with..." or "...allow..." 

I replaced it with allow.

Comment 9: ·      536: repeated sentence

It has been corrected.

Comment 10: 589 missing word: "...high levels of ??? motivation."

The missing word has been added: Identified

Comment 11: ·      629 missing word: 64 years and 4 months

It has been corrected.

Comment 12: ·      consider changing "stress with..." to either "stress from..." or "stress caused by..." or "stress stemming from..." (stress with managers can mean that managers feel stressed)

I replaced it with “stress caused by”.

In closing, we would like to thank you again for your comments. We hope that we have dealt with your suggestions satisfactorily and made all the adjustments requested, both in form and substance.

Yours sincerely,

On behalf of my co-authors,

References added to the manuscript:

References added to the manuscript:

(Issever et al., 2008) Issever Hallim, Kursat Ozdilli, Oguz Altunkaynak, Levent Onen, Rian Disci. 2008. Depression in Tax Office Workers in Istanbul and its Affecting Factors. Indoor and Built Environment, 17(5),414-420. doi:10.1177/1420326X08096609

(Kim, 2018) Kim, Jungin. 2018. The contrary effects of intrinsic and extrinsic motivations on burnout and turnover intention in the public sector," International Journal of Manpower, Emerald Group Publishing Limited, 39(3), 486-500 DOI: 10.1108/IJM-03-2017-0053